# Capturing substructure interactions by invariant Information Bottleneck Theory for Generalizable Property Prediction

## Abstract

Molecular interactions are a common phenomenon in physical chemistry, often resulting in unexpected biochemical properties harmful to humans, such as drug-drug interactions. Machine learning has shown great potential for predicting these interactions rapidly and accurately. However, the complexity of molecular structures and the diversity of interactions often reduce prediction accuracy and hinder generalizability. Identifying core invariant substructures (i.e., rationales) has become essential to improving the model's interpretability and generalization. Despite significant progress, existing models frequently overlook the pairwise molecular interaction, leading to insufficient capture of interaction dynamics. To address these limitations, we propose I2Mole (Interaction-aware Invariant Molecular learning), a novel framework for generalizable property prediction. I2Mole meticulously models atomic interactions, such as hydrogen bonds and Van der Waals forces, by first establishing indiscriminate connections between intermolecular atoms, which are then refined using an improved graph information bottleneck theory tailored for merged graphs. To further enhance model generalization, we construct an environment codebook by environment subgraph of the merged graph. This approach not only could provide noise source for optimizing mutual information but also preserve the integrity of chemical semantic information. By comprehensively leveraging the information inherent in the merged graph, our model accurately captures core substructures and significantly enhances generalization capabilities. Extensive experimental validation demonstrates I2Mole's efficacy and generalizability. The implementation code is available at https://anonymous.4open/r/I2Mol-C616.

## 1 Introduction

The molecular interaction process can generate additional physical or chemical properties when two or more molecules combined Varghese & Mushrif (2019); D'Souza et al. (2011); Low et al. (2022). This phenomenon is common in the fields of physics, chemistry, and medicine *etc.*, such as changes in Gibbs free energy during dissolution (*i.e.*, solute-solvent pair) Chung et al. (2022); Fang et al. (2024); Xia et al. (2023) and synergistic or adverse reactions between drugs (*i.e.*, drug-drug pair) Lee et al. (2023b); Klemperer (1992). Due to the complexity of molecular structures and the diversity of molecular interactions, conventional modeling approaches are limited and susceptible to noise, undermining prediction accuracy. Meanwhile, lacks generalizability and reliability severely limits their applicability. Based on this, mining the invariant core substructures of molecules (*i.e.*, rationale) has gradually become a consensus to enhance both interpretability and generalization like the CGIB Lee et al. (2023a) and MoleOOD Yang et al. (2022b).

Although current methods have attracted widespread attention in predicting the properties of molecular pairs, two inherent shortcomings remain underexplored. The first is **Insufficiency in molecular interaction modeling.** Existing methods demonstrate proficiency in elucidating essential structural characteristics for individual molecular models. However, when molecular interactions occur, pivotal substructures may exhibit considerable variation. Taking Pseudoephedrine (PSE) and Methamphetamine (Meth) as an example (Figure 1 (a)). Both are common drugs used to induce cardiac stimulation, arising from their distinct functional groups affecting neurons. The hydroxyl moiety in PSE participates in regulating neurotransmitter release and reuptake, while the amphetamine moiety

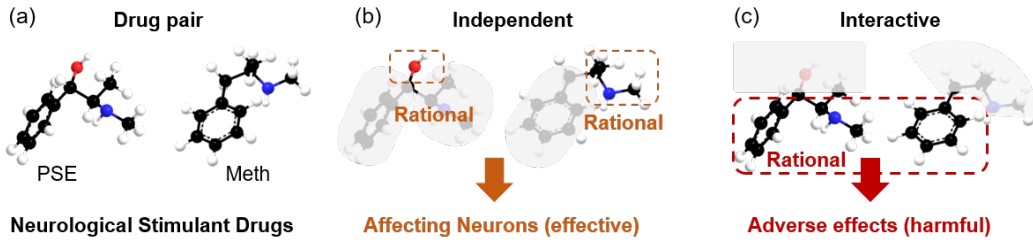

Figure 1: An Example of molecular interactions. (a) Both Pseudoephedrine (PSE) and Methamphetamine (Meth) are stimulant drugs; (b) each drug is influenced by distinct core substructures to stimulate neurons; (c) harmful effects on human health occor by other substructure interaction.

in Meth does this as illustrated in Figure 1 (b). However, upon their concomitant usage, the interaction between Meth's benzene ring and PSE's amphetamine moiety excessively enhances the stimulatory effects on cardiovascular systems, thereby leading to the occurrence of adverse effects and harm to the human body, as shown in Figure 1 (c). Therefore, comprehensive modeling of intermolecular interactions is crucial and necessary for a profound understanding of molecular interactions.

Some current models have noticed the aforementioned shortcomings Behler (2015; 2016), however, they still **lack of consideration for model generalization.** Given the diverse and complex nature of molecular species in real-world scenarios, the data used for training and testing may inevitably be sampled from different distributions, thus presenting challenges related to OOD Paul et al. (2021); Petrova (2013); Yang et al. (2022b). While introducing integrated noise injection techniques to simulate diverse environmental distributions holds promise for enhancing model generalization and capturing core rationals, several drawbacks exist. Specifically, 1) The simulation of noise data may not accurately reflect authentic environmental vectors in chemical space. 2) Indiscriminate application of noise injection can distort semantic information and hinder model convergence. And random environmental vectors may struggle to adequately represent the broad distribution of molecular interactions. 3) When the injected noise variance is minimal, this may converge to a constant, defeating the purpose of its introduction. These limitations compromise the modeling process and may result in suboptimal result.

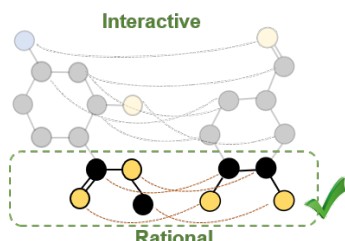

Figure 2: Diagram illustrating molecular interaction modeling to capture rationals. Molecular pairs will be constructed into a merged graph by connecting atoms pairwise (dashed lines). Please note that to avoid excessive complexity, some unimportant relation edges will be removed (unconnected). Best view in color.

In sight of this, we introduce an **I**nteraction-aware **I**nvariant **Mole**cular learning framework, termed I2Mole, for generalizable property prediction. Spontaneous molecular interaction phenomena, such as van der Waals forces Chi et al. (2010), hydrogen bonding interactions Chen et al. (2021), *etc.* are more likely to occur in specific molecular structures (*e.g.*, -OH, =O, N), resulting in stronger forces. We carefully design dynamic weighted relationship edges to differentially learn the interactions between atoms. Conversely, for atomic pairwise interactions that are rarely occur, we employ iterative truncation to restrict their message passing processes, reducing interference with the overall learning of the merged graph while moderately reducing graph complexity, as presented in Figure 2. Given the vast and largely unknown nature of the potential chemical space, collecting all possible environments is impractical. Therefore, we introduce the concept of vector quantization (VQ) van den Oord et al. (2017); Razavi et al. (2019) for molecular interactions to create a merged graphic environment codebook. This codebook clusters the potential environments of molecules in the training set into a specified number of categories. The learned environmental distribution also serves as noise for optimizing mutual information Duncan (1970); Yu et al. (2022b). Therefore, our I2Mole with explict molecule interactions and improved environment codebook, could effectively achieve generalizable property prediction on various DDI datasets.

## 2 PRELIMINARIES

In this section, we begin with a formal description of the problem formulation. Next, we delve into the concepts of the Graph Information Bottleneck and invariant learning.

### 2.1 PROBLEM FORMULATION

A molecule can be depicted as a graph $\mathcal{G}$ whose nodes $\mathcal{V}$ denote the atoms and edges $\mathcal{E}$ act as the bonds Wen et al. (2021). $\mathcal{U}$ is the global feature vector which is extracted from each molecule (Appendix C). Given a set of drug molecular graph pairs $\mathcal{D} = \{(\mathcal{G}_a^1, \mathcal{G}_b^1), (\mathcal{G}_a^2, \mathcal{G}_b^2), \ldots, (\mathcal{G}_a^n, \mathcal{G}_b^n)\}$ and their associated target values $\mathbb{Y} = \{\mathbf{Y}^1, \mathbf{Y}^2, \ldots, \mathbf{Y}^n\}$, our objective is to train a model $\mathcal{M}$ can classify the target values for arbitrary drug pairs in an end-to-end manner, *i.e.*, $\mathbf{Y}^i = \mathcal{M}(\mathcal{G}_a^i, \mathcal{G}_b^i)$.

### 2.2 GRAPH INFORMATION BOTTLENECK (GIB)

According to the GIB principle Yu et al. (2020; 2022b); Miao et al. (2022), we could get:

$$\mathcal{G}_{\text{IB}} = \arg\min_{\mathcal{G}_{\text{sub}} \in \mathcal{S}} -I(\mathbf{Y}; \mathcal{G}_{\text{sub}}) + \beta I(\mathcal{G}; \mathcal{G}_{\text{sub}}). \tag{1}$$

Intuitively, where $\mathcal{S}$ represents the set of $\mathcal{G}_{\text{sub}}$, and $\mathcal{G}_{\text{IB}}$ is the core subgraph of $\mathcal{G}$, which discards information from the $\mathcal{G}$ by minimizing the mutual information $I(\mathcal{G}; \mathcal{G}_{\text{sub}})$, while preserving target-relevant information by maximizing the mutual information $I(\mathbf{Y}; \mathcal{G}_{\text{sub}})$.

### 2.3 INVATIANT LEARNING

Given the distribution shift between training and testing data, recent studies Rojas-Carulla et al. (2018a); Arjovsky et al. (2019); Wu et al. (2022a) propose the existence of a potential environment variable **env** to express this problem:

$$\min_f \max_{\mathcal{G}_{\text{env}} \in \mathbf{E}} \mathbb{E}_{(\mathcal{G}, Y) \sim p(\mathcal{G}, \mathbf{Y}|\mathbf{env} = \mathcal{G}_{\text{env}})} [R(f(\mathcal{G}), \mathbf{Y}) \mid \mathcal{G}_{\text{env}}], \tag{2}$$

where $\mathbf{E}$ denotes the support of environments, $f(\cdot)$ represents the prediction model, and $R(\cdot, \cdot)$ is a risk function. The label $\mathbf{Y}$ is independent of the environment $\mathcal{G}_{\text{env}}$, given the subgraph $\mathcal{G}_{\text{sub}}$:

$$\mathbf{Y} \perp \mathcal{G}_{\text{env}} \mid \mathcal{G}_{\text{sub}}, \tag{3}$$

where $\perp$ denotes probabilistic independence. These principles collectively protect predictions from external influences, ensuring that the rationale comprehensively captures all discriminative features. This is for a single molecule, and we would extend it to molecular pairs.

## 3 METHODOLOGY

In this section, we detail our proposed method. In Section 3.1, we define the merged graph and the intermolecular message passing mechanism. Section 3.2 explains the details of subgraph extraction by GIB theory. In Section 3.3, we describe how injecting environmental embeddings into the rationals to enhance model generalization. Section 3.4 presents the total loss function of I2Mole.

### 3.1 MERGED MOLECULAR REPRESENTATION

**Molecule Merging.** The merged graph $\widetilde{\mathcal{G}}$ could be generated by establishing a weighted relational edge between two molecules which connects each atom pairwise.

$$\widetilde{\mathcal{G}} = \{\mathcal{R}, \mathcal{E}, \mathcal{V}, \mathcal{U}\}. \tag{4}$$

The set of relation edges are $\mathcal{R}$ :

$$\mathcal{R} = \{(r_{ij}, v_{ai}, v_{bj})\}_{k=1}^{N^a \times N^b}, \tag{5}$$

Figure 3: Overview of our model. Initially, molecular pairs construct a merged graph to facilitate the message passing process. Subsequently, subgraphs are extracted based on the GIB, and the environmental components are recorded in a codebook. During the invariant learning process, the rational part concatenates different environment embeddings to achieve invariant representations.

where $ai \in \{1, 2, 3, \ldots, N^a\}$, $bj \in \{1, 2, 3, \ldots, N^b\}$. $N^a$ and $N^b$ is the total number of atoms in drug molecular graph $\mathcal{G}_a$, $\mathcal{G}_b$. $r_{ij}$ represents the relation edge. And $k$ is the index of $\mathcal{R}$.

**Intra-molecular message passing.** Generally, in this merged molecular graph $\widetilde{\mathcal{G}}$, the message passing process is firstly executed intramolecule. In this process, $e_{ij}$ is updated to $e'_{ij}$ by aggregating the initially bond features, as well as the two atomic features, $v_i$ and $v_j$, and the global features $u$. In addition, the atomic feature vector $v_i$ is updated into $v'_i$ and $u$ is updated into $u'_i$:

$$e'_{ij} = e_{ij} + \text{LeakyReLU}[\text{FC}(v_i + v_j) + [\text{FC}(e_{ij}) + [\text{FC}(u)]], \quad (6)$$

$$\hat{e}_{ij} = \frac{\sigma(e'_{ij})}{\sum_{j' \in N_i} \sigma(e'_{ij'}) + \epsilon}, \quad v'_i = v_i + \text{LeakyReLU}[\text{FC}(v_i + \sum_{j \in N_i} \hat{e}_{ij} \odot \text{FC}(v_j)) + \text{FC}(u)], \quad (7)$$

$$u' = u + \text{LeakyReLU}[\text{FC}(\frac{1}{N^v} \sum_{i=1}^{N^v} v'_i + \frac{1}{N^e} \sum_{k=1}^{N^e} e'_k + u)], \quad (8)$$

where FC is a fully connected layer. $\odot$ denotes the Hadamard product. $\sigma(\cdot)$ is the sigmoid activation function, and $\epsilon$ is a fixed constant (0.0001) added for numerical stability. $N^v$ and $N^e$ are the number of atoms and bonds in the molecule, respectively.

**Intermolecular message passing.** We utilize GAT network Veličković et al. (2017) for intermolecular message passing to calculate the weight of relation edge $r_{ij}$.

$$r_{ij} = \text{LeakyReLU}(\text{FC}(\mathbf{W}v'_{ai}, \mathbf{W}v'_{bj})), \quad (9)$$

where $W$ is the learnable weight matrix. According to the calculated attention coefficients ($r_{ij}$) for relation edge, we perform global sorting and retain the top_x% (hyperparameters):

$$r'_{ij} = \begin{cases} r_{ij} & \text{if } r_{ij} \geq X, \\ 0 & \text{otherwise.} \end{cases} \quad (10)$$

Here, $X$ represents the threshold corresponding to the top_x% ranking of $r_{ij}$ values. The selected attention coefficients are then normalized across the entire graph to facilitate the intermolecular information-passing process. The atomic feature $v'_{ai}$ for $i$ is updated, also for $v'_{bj}$:

$$\alpha_{ij} = \frac{r'_{ij}}{\sum_{i,j} r'_{ij}}, \quad v''_{ai} = (1 - \sum_{j \in N_b} \alpha_{ij})v'_{ai} + \sum_{j \in N_b} \alpha_{ij}v'_{bj}, \quad (11)$$

## 3.2 CORE SUBSTRUCTURE EXTRACTION BASED ON GIB

To detect the core reaction structure in merged graph, we optimize the model with the objective function defined in equation 12 as follows:

$$\widetilde{\mathcal{G}}_{\mathrm{IB}} = \arg\min_{\widetilde{\mathcal{G}}_{\mathrm{sub}} \in \widetilde{\mathcal{S}}} - I(\mathbf{Y}; \widetilde{\mathcal{G}}_{\mathrm{sub}}) + \beta I(\mathcal{G}; \widetilde{\mathcal{G}}_{\mathrm{sub}}), \tag{12}$$

where $\widetilde{\mathcal{S}}$ represents the set of $\widetilde{\mathcal{G}}_{\mathrm{sub}}$. Each term indicates the prediction and compression terms respectively, which should be minimized during training, as outlined below.

### 3.2.1 EXTRACT TARGET-ORIENTED INFORMATION

Minimizing $-I(\mathbf{Y}; \widetilde{\mathcal{G}}_{\mathrm{IB}})$, that is to calculate upper bound of $-I(\mathbf{Y}; \widetilde{\mathcal{G}}_{\mathrm{IB}})$. Given the merged graph $\widetilde{\mathcal{G}}$, its label information $\mathbf{Y}$, and the learned IB-graph $\widetilde{\mathcal{G}}_{IB}$, we have:

$$-I(\mathbf{Y}; \widetilde{\mathcal{G}}_{\mathrm{IB}}) \leq \mathbb{E}_{\mathbf{Y}; \widetilde{\mathcal{G}}_{\mathrm{IB}}}[-\log p_\theta(\mathbf{Y}|\widetilde{\mathcal{G}}_{\mathrm{IB}})] := \mathcal{L}_{pre}, \tag{13}$$

where $p_\theta(\mathbf{Y}|\widetilde{\mathcal{G}}_{\mathrm{IB}})$ is variational approximation of $p(\mathbf{Y}|\widetilde{\mathcal{G}}_{\mathrm{IB}})$. We model $p_\theta(\mathbf{Y}|\widetilde{\mathcal{G}}_{\mathrm{IB}})$ as a predictor parametrized by $\theta$. Thus, we can minimize the upper bound of $-I(\mathbf{Y}; \widetilde{\mathcal{G}}_{\mathrm{IB}})$ by minimizing the model prediction loss $\mathcal{L}_{\mathrm{pre}}(\mathbf{Y}, \widetilde{\mathcal{G}}_{\mathrm{IB}})$ with cross entropy loss. More details are given in Appendix B.1 (Sufficiency assumption Yang et al. (2022b)).

### 3.2.2 OPTIMIZE MINIMIZED $\widetilde{G}$

Minimizing $I(\widetilde{\mathcal{G}}; \widetilde{\mathcal{G}}_{\mathrm{IB}})$, that is to calculate upper bound of $I(\widetilde{\mathcal{G}}; \widetilde{\mathcal{G}}_{\mathrm{IB}})$. Inspired by a recent approach on graph information bottleneck Yu et al. (2022b), we also minimizes $I(\widetilde{\mathcal{G}}; \widetilde{\mathcal{G}}_{\mathrm{IB}})$ by injecting noise into node representations. Then we damp the information in $\widetilde{\mathcal{G}}$ by injecting noises into node representations with a learned probability. Let $\epsilon$ be the noise sampled from a parametric noise distribution. We assign each node a probability of being replaced by $\epsilon$. Specifically, for the $i$-th node, the $k$-th relation edge, we learn the probability $p_i$ and $p_k$ with a fully connected layer. Then, we add a Sigmoid function on the output of fully connected layer to ensure $p_i, p_k \in [0, 1]$:

$$p_i = \mathrm{Sigmoid}(\mathrm{FC}(h_i)), \quad p_k = \mathrm{Sigmoid}(\mathrm{FC}(\mathrm{r_k})). \tag{14}$$

Next, if an $k$-th relation edge connected to $i$-th node, we adjust the probability $p_i$ by adding $\frac{p_k}{N}$, where $N$ depends on whether $i$-th node is in $\mathcal{G}_a$ or $\mathcal{G}_b$:

$$p_i = \begin{cases} p_i + \frac{p_k}{N_b} & \text{if } i \in \mathcal{G}_a \text{ and } k\text{-th edge is connected to } i\text{-th node,} \\ p_i + \frac{p_k}{N_a} & \text{if } i \in \mathcal{G}_b \text{ and } k\text{-th edge is connected to } i\text{-th node.} \end{cases} \tag{15}$$

We then replace the node representation $h_i$ by $\epsilon$ with probability $p_i$:

$$z_i = \lambda_i h_i + (1 - \lambda_i)\epsilon, \quad \mathbf{h}_i^r = (1 - \lambda_i)\mathbf{h}_i, \tag{16}$$

where $\lambda_i \sim \mathrm{Bernoulli}(p_i)$, $\mathbf{h}_i^r$ is the irrelevant substructure node which would be used to construct $\widetilde{\mathcal{G}}_{\mathrm{env}}$. The transmission probability $p_i$ controls the information sent from $h_i$ to $z_i$. If $p_i = 1$, then all the information in $h_i$ are transfered to $z_i$ without loss. On the contrary, when $p_i = 0$, then $z_i$ contains no information from $h_i$ but only noise. We hope $p_i$ is learnable so that we can selectively preserve the information in $\widetilde{\mathcal{G}}_{\mathrm{IB}}$. However, $\lambda_i$ is a discrete random variable and we can not directly calculate the gradient of $p_i$. Therefore, we employ the concrete relaxation Jang et al. (2016) for $\lambda_i$:

$$\lambda_i = \mathrm{Sigmoid}(\frac{1}{t} \log \frac{p_i}{1 - p_i} + \log \frac{u}{1 - u}), \tag{17}$$

where $t$ is the temperature parameter and $u \sim \mathrm{Uniform}(0, 1)$. Another critical aspect of noise injection is the characterization of the injected noise. It is important that arbitrary noise can be

detrimental to the semantic integrity of the input graph, leading to predictions that deviate from the actual graph properties. Conversely, appropriately selected noise can provide a variational upper bound to the overall objective. Therefore, the minimizied the upper bound of $I(\widetilde{\mathcal{G}}_{\text{IB}}; \widetilde{\mathcal{G}})$ as follows:

$$I(\widetilde{\mathcal{G}}_{\text{IB}}; \widetilde{\mathcal{G}}) \leq \mathbb{E}_{\mathcal{G}}\left[-\frac{1}{2}\log A_{\widetilde{\mathcal{G}}} + \frac{1}{2m_{\widetilde{\mathcal{G}}}}A_{\widetilde{\mathcal{G}}} + \frac{1}{2m_{\widetilde{\mathcal{G}}}}B_{\widetilde{\mathcal{G}}}^2\right] := \mathcal{L}_{\text{MI}}(\widetilde{\mathcal{G}}_{\text{IB}}, \widetilde{\mathcal{G}}), \tag{18}$$

where $A_{\widetilde{\mathcal{G}}} = \sum_{j=1}^{m_{\tilde{\mathcal{G}}}}(1-\lambda_j)^2$ and $B_{\widetilde{G}} = \frac{\sum_{j=1}^{m_{\tilde{\mathcal{G}}}}\lambda_j(h_j-\mu_h)}{\sigma_h}$. More details are given in Appendix B.2.

## 3.3 ENVIRONMENT INFERENCE

Based on the tailored GIB theory, we can identify the decisive core substructure $\widetilde{\mathcal{G}}_{\text{IB}}$, as present in equation 12. $\widetilde{\mathcal{G}}_{\text{IB}}$ only contains itself information do not guarantee good generalization. Therefore, we are based on the invariance learning theory to enhance the robustness of core substructure by incorporating various environments features captured across downstream tasks. This problem can be formally defined as follows:

$$\min_f \max_{\widetilde{\mathcal{G}}_{\text{env}} \in \mathbf{E}} \mathbb{E}_{(\widetilde{\mathcal{G}}, \mathbf{Y}) \sim q(\widetilde{\mathcal{G}}_{\text{env}})}[R(f(\widetilde{\mathcal{G}}), \mathbf{Y}) \mid \widetilde{\mathcal{G}}_{\text{env}}], \tag{19}$$

where $\mathbf{E}$ denotes the support of environments. The irrelevant substructures $\widetilde{\mathcal{G}}_{\text{env}}$ can be viewed as the environment, where each node embedding is $\mathbf{h}_i^r$. $q(\widetilde{\mathcal{G}}_{\text{env}})$ is the distribution of data under environment $\widetilde{\mathcal{G}}_{\text{env}}$ combined with various rationales, $f(\cdot)$ is the prediction model and $R(\cdot, \cdot)$ is the risk function such as cross-entropy loss. The above equation 19 aims to minimize the maximum errors under different environments thus providing the guarantee of capturing the invariance across environments, which has been well demonstrated in existing researches Wu et al. (2022c;b).

Directly solving the equation 19 is impractical as limited training data across the various environments in $\mathbf{E}$, while the model should be expected to perform well across all these environments. Here, we introduce the concept of vector quantization (VQ) van den Oord et al. (2017); Razavi et al. (2019) to create a trainable environment codebook $W = \{env_1, env_2, \ldots, env_M\}$, defining a latent embedding space $env \in \mathbb{R}^{M \times F}$. Here, $M$ represents the number of discrete environments (*i.e. env*), and $F$ denotes the dimension of each latent vector. Then, we use a nearest neighbor lookup in the shared embedding space $\mathbf{E}$ to find the closest latent vector $env_m$, indexed by $m$.

Additionally, the set2set network Vinyals et al. (2015) is utilized to pool $\widetilde{\mathcal{G}}_{\text{IB}}, \widetilde{\mathcal{G}}_{\text{env}}, \widetilde{\mathcal{G}}$, resulting in the substructure representation vectors $\widetilde{s}_{\text{IB}}, \widetilde{s}_{\text{env}}$ and $\widetilde{s}_{\mathcal{G}}$. This process acts as a specific non-linearity that maps the latent vectors $\widetilde{s}_{\text{env}}$ to one of the $M$ embedding vectors:

$$q\left(m \mid \widetilde{s}_{env}\right) = \begin{cases} 1 & \text{for } m = \arg\min_j \|\widetilde{s}_{\text{env}} - env_j\|_2, \\ 0 & \text{otherwise.} \end{cases} \tag{20}$$

Aiming to update the codebook and encourage the output of the encoder to stay close to the chosen codebook embedding, we defined $\mathcal{L}_{vq}$ as follow, where the sg$[\cdot]$ denotes stop-gradient and $\delta$ is a hyper-parameter set to 0.25 Xia et al. (2022):

$$\mathcal{L}_{vq} = \|\text{sg}[\widetilde{s}_{\text{env}}] - env_m\|_2^2 + \delta\|\widetilde{s}_{\text{env}} - \text{sg}[env_m]\|_2^2. \tag{21}$$

As $\mathcal{L}_{vq}$ gradually converges, we obtain a stable codebook set $W$, which clusters the infinite possible environment space $\mathbf{E}$ into a discretized set of $M$ finite environments represented by $W$. Subsequently, we traverse all potential environment vectors ($env$) and assign rationals to different environments for achieving stable predictions. This ensures that the prediction results of the rationals are independent of the environments, thereby guaranteeing the independence between the rationals and the environment samples (Invariance assumption Yang et al. (2022b)).

Table 1: Performance of different methods in transudative setting. (Bold numbers are the best results, while the top-performing baseline is highlighted with a superscript cross).

| | ZhangDDI | | | ChchMiner | | | DeepDDI | | |
|---|---|---|---|---|---|---|---|---|---|
| | ACC (↑) | AUROC (↑) | F1 (↑) | ACC (↑) | AUROC (↑) | F1 (↑) | ACC (↑) | AUROC (↑) | F1 (↑) |
| GoGNN | $84.14_{(0.46)}$ | $92.35_{(0.48)}$ | $81.54_{(0.42)}$ | $91.17_{(0.46)}$ | $96.64_{(0.40)}$ | $92.35_{(0.34)}$ | $93.54_{(0.35)}$ | $92.71_{(0.27)}$ | $89.83_{(0.41)}$ |
| DeepDDI | $83.35_{(0.49)}$ | $91.13_{(0.58)}$ | $80.24_{(0.47)}$ | $90.34_{(0.44)}$ | $95.71_{(0.37)}$ | $91.83_{(0.28)}$ | $92.39_{(0.39)}$ | $98.10_{(0.42)}$ | $91.32_{(0.39)}$ |
| SSI-DDI | $86.97_{(0.62)}$ | $93.76_{(0.34)}$ | $82.99_{(0.30)}$ | $93.26_{(0.24)}$ | $97.81_{(0.22)}$ | $93.11_{(0.19)}$ | $94.27_{(0.25)}$ | $97.42_{(0.31)}$ | $95.41_{(0.19)}$ |
| MHCADDI | $77.86_{(0.50)}$ | $86.94_{(0.68)}$ | $83.67_{(0.48)}$ | $84.26_{(0.64)}$ | $89.33_{(0.72)}$ | $83.21_{(0.53)}$ | $87.01_{(0.77)}$ | $97.64_{(0.83)}$ | $88.54_{(0.55)}$ |
| MDF-SA-DDI | $86.89_{(0.15)}$ | $94.03_{(0.22)}$ | $83.67_{(0.14)}$ | $94.63_{(0.21)}†$ | $98.10_{(0.17)}$ | $94.17_{(0.16)}$ | $94.12_{(0.21)}$ | $88.84_{(0.26)}$ | $96.13_{(0.17)}$ |
| DSN-DDI | $87.65_{(0.13)}$ | $94.63_{(0.18)}†$ | $84.30_{(0.09)}$ | $94.25_{(0.11)}$ | $98.31_{(0.10)}$ | $95.34_{(0.08)}$ | $95.74_{(0.18)}$ | $98.06_{(0.16)}$ | $96.71_{(0.11)}$ |
| CGIB | $87.32_{(0.71)}$ | $94.43_{(0.60)}$ | $84.53_{(0.45)}$ | $94.37_{(0.39)}$ | $98.38_{(0.31)}†$ | $95.44_{(0.24)}$ | $95.76_{(0.72)}†$ | $98.08_{(0.64)}†$ | $96.53_{(0.53)}$ |
| CMRL | $87.78_{(0.37)}†$ | $94.08_{(0.23)}$ | $84.78_{(0.25)}†$ | $94.43_{(0.25)}$ | $98.37_{(0.12)}$ | $95.62_{(0.17)}†$ | $95.49_{(0.34)}$ | $98.03_{(0.31)}$ | $96.82_{(0.29)}†$ |
| Ours | $\mathbf{88.64}_{(0.24)}$ | $\mathbf{95.12}_{(0.12)}$ | $\mathbf{85.87}_{(0.20)}$ | $\mathbf{95.34}_{(0.19)}$ | $\mathbf{98.84}_{(0.10)}$ | $\mathbf{96.21}_{(0.25)}$ | $\mathbf{96.51}_{(0.14)}$ | $\mathbf{99.04}_{(0.22)}$ | $\mathbf{97.53}_{(0.15)}$ |

$$\min_f \mathbb{E}_{env_i \in W} \mathbb{E}_{(\widetilde{s}_\mathcal{G}, \mathbf{Y}) \sim q(env_i)} \left[ R\left( f\left( \widetilde{s}_\mathcal{G} \right), \mathbf{Y} \right) \mid env_i \right]. \tag{22}$$

This formula can be obtained by minimizing the weighted sum of cross-entropy losses across different environments. Assuming a total of $C$ classes, let $\phi_i$ denote the probability that $env$ belongs to $env_i$, and $\Phi$ represents the classification head that maps the molecular representation to the category labels. the encoder $f_{env}$ and the classification head $\Phi$ together form the prediction model. Consequently, the loss can be expressed in the following form, where ∥ denotes the concatenation operation:

$$\mathcal{L}_{inv} = -\sum_{i=1}^{M} \phi_i \sum_{\widetilde{s}_\mathcal{G} \in \mathcal{D}_{\text{train}}} \sum_{c=1}^{C} \mathbf{Y}_{\widetilde{s}_\mathcal{G}} \log \Phi\left( f_{env}(\widetilde{s}_{\text{IB}} \| env_i) \right). \tag{23}$$

## 3.4 Training Objective

Finally, we train the model with the following objective:

$$\mathcal{L}_{\text{total}} = \mathcal{L}_{\text{inv}} + \mathcal{L}_{\text{pre}} + \beta \mathcal{L}_{\text{MI}} + \gamma \mathcal{L}_{\text{vq}} \tag{24}$$

Here, $\mathcal{L}_{\text{pre}}$ and $\mathcal{L}_{\text{MI}}$, guided by the GIB. $\mathcal{L}_{\text{pre}}$ is the cross-entropy loss for classification tasks. $\mathcal{L}_{\text{MI}}$ represents the KL divergence between the core substructures and the non-core subgraph, encouraging substructure compression. And $\mathcal{L}_{\text{inv}}$ aims to minimizing the disturbance loss across various environments. $\beta$ and $\gamma$ are trade-off parameters which governs the weight of $\mathcal{L}_{\text{MI}}$ and $\mathcal{L}_{\text{vq}}$.

## 4 Experiment and Analyse

We present experimental results to demonstrate the effectiveness of I2Mole. In this section, we conduct extensive experiments to answer the following questions:

- **RQ1:** Can I2Mole improve the prediction accuracy on various drug-drug pairs?
- **RQ2:** How effective is I2Mole in terms of generalization capabilities?
- **RQ3:** What does the environment codebook represent and learned in the model?

### 4.1 Datasets and setups

**Datasets.** To evaluate the performance of our model, we conduct experiments based on three commonly used datasets in DDI event prediction task, including ZhangDDI Zhang et al. (2017), DeepDDI Ryu et al. (2018) and ChChMiner Zitnik et al.. More detailed description could be found in Appendix D.1.

**Baselines.** In our extensive assessment, our model is compared with eight advanced DDI event prediction methods, all leveraging molecular graphs as input features. The compared methods include GoGNN Wang et al. (2020a), MHCADDI Deac et al. (2019), DeepDDI Ryu et al. (2018), SSI-DDI

Table 2: Performance of different methods in inductive settings. (Bold numbers are the best results, while the top-performing baseline is highlighted with a superscript cross).

| | Type1 | | | | | | | | |
|---|---|---|---|---|---|---|---|---|---|
| | **ZhangDDI** | | | **ChchMiner** | | | **DeepDDI** | | |
| | **ACC** (↑) | **AUROC** (↑) | **F1** (↑) | **ACC** (↑) | **AUROC** (↑) | **F1** (↑) | **ACC** (↑) | **AUROC** (↑) | **F1** (↑) |
| GoGNN | $61.51_{(1.87)}$ | $63.17_{(1.42)}$ | $45.53_{(1.28)}$ | $67.48_{(1.56)}$ | $69.52_{(1.84)}$ | $69.22_{(1.33)}$ | $67.53_{(1.52)}$ | $71.34_{(1.24)}$ | $67.16_{(1.13)}$ |
| DeepDDI | $60.84_{(1.34)}$ | $59.51_{(1.18)}$ | $43.81_{(1.26)}$ | $66.19_{(1.08)}$ | $68.51_{(1.53)}$ | $67.67_{(1.29)}$ | $64.39_{(1.71)}$ | $69.52_{(1.53)}$ | $68.31_{(1.45)}$ |
| SSI-DDI | $62.38_{(1.53)}$ | $69.56_{(1.21)}$ | $47.59_{(1.17)}$ | $76.94_{(1.32)}$ | $79.64_{(1.53)}$ | $77.61_{(1.24)}$ | $69.77_{(0.86)}$ | $75.93_{(1.14)}$ | $72.23_{(0.77)}$ |
| MHCADDI | $61.81_{(1.27)}$ | $67.52_{(0.97)}$ | $44.51_{(1.38)}$ | $72.77_{(0.76)}$ | $70.92_{(1.08)}$ | $78.15_{(0.97)}$ | $65.94_{(0.98)}$ | $72.18_{(0.87)}$ | $70.37_{(1.24)}$ |
| MDF-SA-DDI | $64.51_{(1.39)}$ | $70.99_{(1.27)}$ | $51.53_{(1.15)}$ | $75.39_{(0.80)}$ | $80.47_{(0.68)}$ | $79.83_{(1.05)}$ | $71.13_{(0.77)}$ | $80.54_{(0.94)}$ | $71.61_{(0.88)}$ |
| DSN-DDI | $67.68_{(0.87)}$ | $72.49_{(1.02)}$ | $53.64_{(0.77)}$ | $78.94_{(0.72)}$ | $85.93_{(0.65)}$ | $83.81_{(0.83)}$ | $73.35_{(0.62)}$ | $83.11_{(0.76)}$ | $75.68_{(0.70)}$ |
| CGIB | $68.34_{(0.66)}$ | $72.80_{(0.43)}$ | $57.29_{(0.58)}$† | $79.75_{(0.73)}$ | $86.41_{(0.93)}$ | $85.13_{(0.43)}$ | $73.86_{(0.97)}$ | $80.80_{(0.53)}$ | $78.47_{(0.47)}$† |
| CMRL | $68.38_{(1.12)}$† | $74.59_{(1.05)}$† | $56.41_{(0.97)}$ | $80.54_{(0.66)}$† | $87.64_{(0.54)}$† | $86.55_{(0.57)}$† | $74.12_{(0.55)}$† | $84.96_{(0.87)}$† | $77.81_{(0.74)}$ |
| Ours | $\mathbf{69.12}_{(0.23)}$ | $\mathbf{75.14}_{(0.42)}$ | $\mathbf{57.89}_{(1.55)}$ | $\mathbf{81.59}_{(1.10)}$ | $\mathbf{88.51}_{(0.31)}$ | $\mathbf{87.43}_{(0.74)}$ | $\mathbf{75.27}_{(0.0.64)}$ | $\mathbf{85.62}_{(0.74)}$ | $\mathbf{78.96}_{(0.37)}$ |

| | Type2 | | | | | | | | |
|---|---|---|---|---|---|---|---|---|---|
| | **ZhangDDI** | | | **ChchMiner** | | | **DeepDDI** | | |
| | **ACC** (↑) | **AUROC** (↑) | **F1** (↑) | **ACC** (↑) | **AUROC** (↑) | **F1** (↑) | **ACC** (↑) | **AUROC** (↑) | **F1** (↑) |
| GoGNN | $55.37_{(3.27)}$ | $54.37_{(2.47)}$ | $34.92_{(3.26)}$ | $64.27_{(4.31)}$ | $67.73_{(3.63)}$ | $72.19_{(4.30)}$ | $62.96_{(3.64)}$ | $63.91_{(3.61)}$ | $68.53_{(3.34)}$ |
| DeepDDI | $58.62_{(2.03)}$ | $56.34_{(1.97)}$ | $25.19_{(4.34)}$ | $63.78_{(2.14)}$ | $66.71_{(2.67)}$ | $71.37_{(3.38)}$ | $61.68_{(4.18)}$ | $65.17_{(3.72)}$ | $66.74_{(4.16)}$ |
| SSI-DDI | $57.24_{(2.38)}$ | $59.34_{(3.26)}$ | $37.16_{(3.84)}$ | $65.61_{(2.51)}$ | $68.39_{(1.94)}$ | $74.95_{(2.17)}$ | $65.53_{(3.53)}$ | $69.37_{(4.16)}$ | $62.18_{(3.94)}$ |
| MHCADDI | $57.84_{(2.28)}$ | $56.47_{(2.77)}$ | $33.53_{(3.18)}$ | $59.24_{(5.39)}$ | $63.57_{(4.47)}$ | $64.57_{(4.17)}$ | $60.17_{(3.67)}$ | $62.89_{(3.42)}$ | $63.57_{(5.17)}$ |
| MDF-SA-DDI | $57.63_{(1.89)}$ | $55.97_{(1.67)}$ | $33.94_{(2.78)}$ | $65.24_{(1.97)}$ | $68.54_{(2.04)}$ | $77.32_{(1.89)}$ | $66.34_{(1.55)}$ | $70.81_{(2.01)}$ | $70.95_{(1.71)}$ |
| DSN-DDI | $58.37_{(1.31)}$ | $58.88_{(1.12)}$ | $39.49_{(2.32)}$ | $68.36_{(1.54)}$ | $69.34_{(1.34)}$ | $77.52_{(1.21)}$ | $68.17_{(1.28)}$ | $72.71_{(1.37)}$ | $71.96_{(1.64)}$ |
| CGIB | $58.39_{(2.04)}$ | $57.24_{(1.97)}$ | $28.83_{(4.53)}$ | $68.78_{(1.84)}$† | $69.82_{(1.39)}$† | $78.46_{(2.03)}$† | $68.26_{(1.39)}$ | $68.78_{(1.67)}$ | $75.75_{(1.75)}$† |
| CMRL | $60.78_{(1.37)}$† | $60.02_{(2.03)}$† | $38.73_{(3.04)}$† | $67.09_{(1.54)}$ | $69.62_{(1.67)}$ | $75.76_{(1.28)}$ | $68.29_{(1.78)}$† | $73.38_{(1.96)}$† | $73.91_{(2.14)}$ |
| Ours | $\mathbf{61.35}_{(1.01)}$ | $\mathbf{62.02}_{(1.09)}$ | $\mathbf{39.95}_{(1.56)}$ | $\mathbf{69.23}_{(0.17)}$ | $\mathbf{70.02}_{(0.85)}$ | $\mathbf{79.67}_{(0.35)}$ | $\mathbf{69.92}_{(0.11)}$ | $\mathbf{74.27}_{(0.62)}$ | $\mathbf{75.83}_{(0.43)}$ |

Nyamabo et al. (2021), CGIB Lee et al. (2023a), CMRL Lee et al. (2023c), MDF-SA-DDI Lin et al. (2022)and DSN-DDI Li et al. (2023). More detailed description could be found in Appendix D.2.

**Metric.** Three metrics are employed to evaluate the model performance: accuracy (ACC), area under the receiver operating characteristic (AUROC), harmonic mean of precision and recall (F1). All experiments are repeated eight times with the same dataset split and average result is presented.

## 4.2 MODEL PERFORMANCE (RQ1)

Similar to previous studiesDeac et al. (2019); Nyamabo et al. (2021); Ryu et al. (2018), we first performed the transductive setting that is the common method evaluation scheme, where the entire dataset is randomly split and aims to predict the undiscovered DDI events among known drugs. In this scenario, we split the dataset into training (60%), validation (20%), and test (20%) parts. Based on the results presented in Table 1, three key observations can be delineated:

**Obs.1: I2Mole exhibits the excited predictive performance in transductive setting.** The results of our model and eight baseline models are documented in Table 1. We observe that our model demonstrates the optimal predictive performance across three different scales of datasets. Regarding the ACC evaluation metric, it outperforms other models on the ZhangDDI and DeepDDI datasets, while its performance on the ChChMiner dataset is comparable to MDF-SA-DDI.

**Obs.2: I2Mole shows more pronounced performance improvements on the large-scale DeepDDI dataset.** The model's performance across different datasets may be influenced by variations in dataset characteristics, where larger datasets imply a greater diversity of drugs and more complex DDI relationships. Compared to the second-best model, I2Mole has improved 1.03% on the large-scale DeepDDI dataset, while only 0.96% on the medium and small-scale datasets in AUROC index.

## 4.3 GENERALIZATION TEST (RQ2)

In this section, we evaluated I2Mole's generalizability by inductive settings and domain shifting test. Type 1 aims to predict potential interaction properties between known and unseen drugs, while Type 2 aims to predict potential interaction properties between unseen and unseen drugs as in Table 2.

**Obs.3: I2Mole demonstrates excellent generalization ability on inductive settings.** We assessed the generalization capability on I2Mole to unseen drugs, which holds significant practical and real-world implications. This process was implemented by partitioning drugs, and the testing results, in

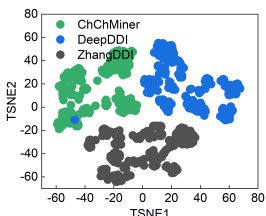

Figure 4: TSNE map for three DDI datasets (3000 drug pairs are respectively selected.)

Table 3: Performance on domain generalization experiments.

| | ChchMiner | | | DeepDDI | | |
|---|---|---|---|---|---|---|
| | ACC ($\uparrow$) | AUROC ($\uparrow$) | F1 ($\uparrow$) | ACC ($\uparrow$) | AUROC ($\uparrow$) | F1 ($\uparrow$) |
| GoGNN | $47.81_{(0.35)}$ | $60.25_{(0.21)}$ | $62.53_{(0.32)}$ | $51.20_{(0.26)}$ | $66.24_{(0.34)}$ | $62.35_{(0.35)}$ |
| DeepDDI | $48.27_{(0.24)}$ | $61.21_{(0.32)}$ | $60.25_{(0.27)}$ | $50.34_{(0.14)}$ | $65.21_{(0.27)}$ | $61.83_{(0.25)}$ |
| SSI-DDI | $51.25_{(0.21)}$ | $60.47_{(0.31)}$ | $62.34_{(0.52)}$ | $53.26_{(0.45)}$ | $67.24_{(0.42)}$ | $63.11_{(0.34)}$ |
| MHCADDI | $52.45_{(0.12)}$ | $62.43_{(0.31)}$ | $63.27_{(0.38)}$ | $54.26_{(0.24)}$ | $67.53_{(0.22)}$ | $63.21_{(0.34)}$ |
| MDF-SA-DDI | $33.54_{(0.12)}$ | $65.34_{(0.32)}$ | $63.55_{(0.54)}$ | $54.63_{(0.34)}$ | $68.50_{(0.25)}$ | $64.17_{(0.21)}$ |
| DSN-DDI | $52.24_{(0.24)}$ | $62.45_{(0.28)}$ | $64.20_{(0.09)}$ | $54.86_{(0.21)}$ | $68.25_{(0.24)}$ | $65.34_{(0.24)}$ |
| CGIB | $55.21_{(0.21)}$ | $67.54_{(0.46)}$ | $64.53_{(0.44)}$ | $55.37_{(0.29)}$ | $68.48_{(0.45)}$ † | $65.44_{(0.32)}$ |
| CMRL | $55.76_{(0.21)}$ † | $68.14_{(0.23)}$ † | $64.82_{(0.15)}$ † | $56.43_{(0.55)}$ † | $68.45_{(0.21)}$ | $65.62_{(0.45)}$ † |
| Ours | $\mathbf{59.25}_{(0.15)}$ | $\mathbf{69.22}_{(0.31)}$ | $\mathbf{65.25}_{(0.26)}$ | $\mathbf{58.12}_{(0.09)}$ | $\mathbf{68.72}_{(0.42)}$ | $\mathbf{65.76}_{(0.23)}$ |

Table 4: The scaffold and size splitting experiments result.

| Model | ZhangDDI | | Chchminer | | DeepDDI | |
|---|---|---|---|---|---|---|
| | Scaffold ($\uparrow$) | Size ($\uparrow$) | Scaffold ($\uparrow$) | Size ($\uparrow$) | Scaffold ($\uparrow$) | Size ($\uparrow$) |
| SSI-DDI | $72.34_{(3.73)}$ | $70.15_{(2.47)}$ | $72.34_{(1.22)}$ | $76.89_{(1.82)}$ | $78.27_{(2.12)}$ | $84.75_{(4.32)}$ |
| MHCADDI | $75.67_{(1.34)}$ | $76.28_{(1.53)}$ | $82.71_{(1.25)}$ | $82.66_{(2.17)}$ | $84.57_{(3.24)}$ | $82.74_{(2.13)}$ |
| MDF-SA-DDI | $79.34_{(1.37)}$ | $79.46_{(0.48)}$ | $85.47_{(0.75)}$ | $84.23_{(0.63)}$ | $87.38_{(0.32)}$ | $86.58_{(0.37)}$ |
| DSN-DDI | $82.16_{(1.21)}$ | $80.38_{(1.23)}$ | $87.47_{(2.14)}$ | $87.92_{(1.32)}$ | $88.99_{(2.33)}$ | $86.53_{(1.65)}$ |
| CGIB | $83.32_{(1.26)}$ | $80.79_{(0.83)}$ | $89.47_{(1.10)}$ | $88.43_{(1.39)}$ | $89.56_{(2.22)}$ | $89.44_{(2.02)}$ |
| CMRL | $82.25_{(0.85)}$ | $81.32_{(0.77)}$ | $89.78_{(1.24)}$ | $88.49_{(1.91)}$ | $90.76_{(2.31)}$ | $90.77_{(1.22)}$ |
| Ours | $\mathbf{83.45}_{(0.92)}$ | $\mathbf{82.55}_{(1.12)}$ | $\mathbf{90.32}_{(1.71)}$ | $\mathbf{89.95}_{(1.06)}$ | $\mathbf{91.76}_{(2.63)}$ | $\mathbf{91.89}_{(1.42)}$ |

comparison with baseline models, are documented in Table 2. Evidently, when predicting with new drugs, the performance of all models experiences varying degrees of decline. But I2Mole exhibiting excellent predictive performance has the minimized sensitivity to unseen drug pairs.

**Obs.4: I2Mole shows robust performance on domain generalization experiments.** To investigate the impact of domain shifting on generalization, we transfer a model trained on a smaller dataset to a larger dataset. Specifically, I2Mole trained on the ZhangDDI dataset will test on two other datasets. Notably, ZhangDDI and DEEPDDI exhibit entirely distinct distributions of molecular species, as depicted in Figure 4. I2Mole outperforms other baseline models consistently across all conditions, underscoring its superior generalization capability, as recoded in Table 3.

**Obs.5: I2Mole demonstrates superior performance in scaffold and size splitting experiments.** As presented in Table 4, our proposed model consistently surpasses state-of-the-art methods, achieving the highest accuracy in both scaffold and size splits. These results underscore the advantages of our approach, enabling the model to effectively extract rationales with impressive generalization capabilities, performing robustly across different test scenarios.

### 4.4 EXPLORING THE IMPACT AND EFFECTS OF ENVIRONMENT CODEBOOK (RQ3)

In this section, we provide an intuitive understanding through t-SNE analysis of environment vectors and molecular embeddings, as presented in Figure 5.

**Obs.6: Different environment embeddings in environment codebook has clear boundaries on visualization result.** The 10 distinct environment embeddings exhibit clear distinctions (Figure 5 (a)) ensure that the model adequately learns different types of environmental variables and thereby enhancing its generalization. Moreover, different molecular substructure embeddings are tightly centered around the corresponding environment embedding, respectively. This suggests that updating the codebook vector is essentially equivalent to performing clustering on the molecular embeddings, with the environment embeddings serving as the clustering centers as shown in Figure 5 (a).

**Obs.7: Different environment codes tend to encode the local environments of different molecular pairs.** Figure 5(b) shows the distribution of atom types for each environmental embedding which is close to real-world data. Notably, significant differences exist between environment codes; for example, carbon is predominant in Category 7, while nitrogen and oxygen play unignorable roles in Category 5 and 6. These environmental embeddings represent the non-core substructures of molecular

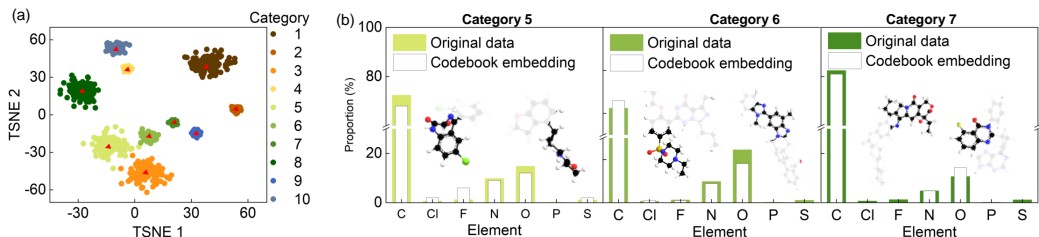

Figure 5: Environmental codebook vectors analysis. (a) TSNE dimensionality reduction plot of drug molecular pairs and the 10-class environment codebook vectors. Different colors represent the chosen codebook vectors, with red dots within clusters indicating the codebook vectors location. (b) Elemental composition of molecular pairs in clusters 5, 6, and 7 (colored) compared to the elemental composition represented by the codebook vectors (blank), along with an example pair of molecules.

Table 5: Ablation experiment. Intermolecular interaction denotes as $\Delta$.

|  | ZhangDDI | | |
| --- | --- | --- | --- |
|  | **ACC** | **AUROC** | **F1** |
| w/o VQ | $74.52_{(0.11)}$ | $83.61_{(0.13)}$ | $74.01_{(0.24)}$ |
| w/o $\Delta$ | $84.51_{(0.22)}$ | $87.21_{(0.27)}$ | $80.21_{(0.33)}$ |
| w/o GIB | $84.72_{(0.08)}$ | $87.21_{(0.24)}$ | $81.07_{(0.43)}$ |
| Ours | $88.02_{(0.24)}$ | $94.73_{(0.12)}$ | $85.07_{(0.20)}$ |

Table 6: Sensitivity analysis for $\beta$ and $\gamma$ (ACC indicator).

|  | ZhangDDI | | | | |
| --- | --- | --- | --- | --- | --- |
|  | **0** | **1E-5** | **1E-4** | **1E-3** | **0.1** |
| $\beta$ | $85.71_{(0.02)}$ | $85.84_{(0.08)}$ | $87.73_{(0.18)}$ | $86.83_{(0.03)}$ | $56.46_{(0.03)}$ |
|  | **2E-5** | **1E-4** | **2E-4** | **5E-4** | **1E-3** |
| $\gamma$ | $88.01_{(0.08)}$ | $88.02_{(0.04)}$ | $88.01_{(0.02)}$ | $88.02_{(0.01)}$ | $88.03_{(0.01)}$ |

pairs. For each codebook category, we provide examples of molecular pairs in Figure 5(b), illustrating the types of real-world substructures represented. Additional visualizations are in Appendix I.

## 5 ABLATION STUDY AND SENSITIVITY ANALYSIS

**Obs.8: Ablation study.** To further investigate the role of each component, we conducted a series of ablation studies. As shown in Table 5, removing the GIB module reduced the model's ability to capture core substructures, limiting its performance. Similarly, the removal of intermolecular interaction disrupted accurate chemical modeling, degrading the model's capabilities. Notably, eliminating VQ module led to substantial performance drops, highlighting the importance of codebook and vector quantization operation (more detailed result could be fould in Appendix F). The computational complexity and data scalability analysis are present in Appendix H.

**Obs.9: Sensitivity analysis.** We investigate the sensitivity of $\beta$ and $\gamma$, which govern the trade-off between prediction and compression, and the codebook updating process, respectively. These parameters correspond to the weights of $\mathcal{L}_{\text{MI}}$ in $\mathcal{L}_{\text{vq}}$ in equation 24. Overall, the model demonstrates robustness to variations in $\beta$ and $\gamma$, but performance degrades significantly when $\beta$ is sharply increased. More detailed sensitivity analysis results, including the number of samples and environmental vectors, among other factors, are presented in Appendix F.

## 6 CONCLUSION AND FUTURE OUTLOOK

In this work, we introduce I2Mole, a novel framework for precise DDI prediction. I2Mole constructs a merged drug pair graph to capture complex molecular interactions, specifically addressing the imbalance between training and testing data distributions commonly encountered in real-world drug scenarios. By leveraging an enhanced information bottleneck approach, I2Mole extracts invariant and rational subgraphs, ensuring consistency across diverse environments. Furthermore, we devise an environment codebook based on the molecular environments in the training set. This captures environmental information and integrates it into data from diverse distributions, thereby significantly enhancing the model's generalization capability. Extensive experiments validate I2Mole's superior accuracy and generalizability, facilitating the rapid identification of potential DDIs and reducing the risks associated with drug misuse. Limitations are discussed in Appendix I.

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

# A    REALTED WORK

## A.1    DRUG-DRUG INTERACTION (DDI) PREDICTION

In recent years, computational approaches, particularly employing machine learning and deep learning methods, have emerged as indispensable tools for swiftly and economically predicting potential DDIs Ryall & Tan (2015); Jaaks et al. (2022). Initially, DDI prediction models predominantly focused on drug attribute information, assuming that similar drugs would exhibit common interactions Ryu et al. (2018); Deng et al. (2020). For instance, Gottlieb et al. (2012) utilized seven types of drug features to construct similarity vectors, forming a DDI prediction model based on logistic regression. Ferdousi et al. (2017) designed a deep neural network using drug molecular similarity vectors as descriptors for predicting potential DDIs. Recently, there has been a shift towards graph-based DDI prediction methodologies. Zhong et al. (2019) employed Graph Convolutional Neural Networks (GC-NNs) for message aggregation and an attention-based pooling method for DDI prediction. Given that the interaction between two drugs is influenced by their specific substructures and functions, recent efforts have focused on substructure extraction and interaction  Harrold & Zavod (2014); Fu et al. (2020). For instance, Yu et al. (2022a) utilized functional group information of drug molecules as their substructures, while Nyamabo et al. Nyamabo et al. (2021) introduced the Substructure-Substructure Interaction for Drug-Drug Interaction (SSI-DDI) method, employing graph attention network (GAT) layers to extract substructure representations and co-attention layers to model interactions among substructures.

Despite the proficiency of existing methodologies in elucidating essential structural characteristics of individual molecular models  Tang et al. (2023); Lee et al. (2023c), considerable variation in crucial substructures may occur during molecular interactions. Notably, while some pioneering work like DSIL-DDI Tang et al. (2023) and CMRL Lee et al. (2023c) has provided foundational insights, a noticeable gap remains in comprehensively modeling intermolecular interactions. CMRL Lee et al. (2023c) innovatively incorporates conditional graph information bottleneck theory to obtain rationales, simultaneously considering a second drug as a conditional factor during drug subgraph generation Lee et al. (2023b). However, prevailing methodologies encounter limitations in adequately capturing molecular interactions, particularly at the atomic level. Moreover, integrating a comprehensive profile of interacting molecules into subgraph generation poses significant challenges, including overwhelming complexity and the risk of incorporating redundant information Jia et al. (2009).

## A.2    OOD GENERALIZATION

The susceptibility of deep neural networks to significant performance degradation under distribution shifts has spurred extensive research on out-of-distribution (OOD) generalization. In response, the invariant rationalization theory has been introduced, aiming to achieve an invariant representation across diverse environments Chang et al. (2020); Rojas-Carulla et al. (2018b). This theory involves a rationalization module that discerns a crucial subset within the input graph, referred to as rationale, essential for prediction Ying et al. (2019); Luo et al. (2020). Subsequently, through invariant learning, these rationales are exposed to diverse environments, thereby fortifying the learned representation against environmental fluctuations and effectively bolstering the model's OOD capacity. 1) **sufficiency:** shows sufficient predictive power for the target, 2) **invariance:** contributes to equal (optimal) performance for the downstream tasks across all environments. Certain methods in computer vision Lv et al. (2022); Zhang et al. (2020); Wang et al. (2020b) achieve OOD generalization by learning domain-invariant representations. Additionally, methods such as Shen et al. (2018); He et al. (2021); Shen et al. (2020) aim to achieve OOD generalization by decorrelating correlated and irrelevant features, considering the statistical correlation between these features as a major factor for distribution shifts. In terms of molecular applications, DIR Wu et al. (2022d) introduces an inventive method to unveil invariant rationales by intervening in the training distribution, generating multiple interventional distributions, and identifying causal rationales consistent across varied distributions. Similarly, MoleOOD Yang et al. (2022a) suggests that leveraging causal data-generating invariance from substructures across environments, linked to specific properties, holds promise for enhancing OOD generalization. However, learning a domain-invariant representation for intermolecular interaction remains an open problem, and current discussions on OOD issues among molecules are limited.

## B PROOFS

### B.1 PROOF OF $\mathcal{L}_{pre}$

*Proof.* Regarding $I(Y; \widetilde{\mathcal{G}})$, we consider $P_\theta\left(Y \mid \widetilde{\mathcal{G}}\right)$ as the variational estimation of $P\left(Y \mid \widetilde{\mathcal{G}}\right)$. Therefore, we can proceed with the following derivation:

$$
\begin{aligned}
I(Y; \widetilde{\mathcal{G}}) &= \mathbb{E}_{(Y, \widetilde{\mathcal{G}})} \log \left[ \frac{P\left(Y \mid \widetilde{\mathcal{G}}\right)}{P(Y)} \right] \\
&= \mathbb{E}_{(Y, \widetilde{\mathcal{G}})} \log \left[ \frac{P_\theta\left(Y \mid \widetilde{\mathcal{G}}\right)}{P(Y)} \right] + \\
&\quad \mathbb{E}_{\widetilde{\mathcal{G}}} \log \left[ KL \left( P\left(Y \mid \widetilde{\mathcal{G}}\right) \| P_\theta\left(Y \mid \widetilde{\mathcal{G}}\right) \right) \right].
\end{aligned}
\tag{25}
$$

Considering the non-negativity property of the Kullback-Leibler divergence, we can conclude that:

$$
\begin{aligned}
I(Y; \widetilde{\mathcal{G}}) &\geq \mathbb{E}_{(Y, \widetilde{\mathcal{G}})} \log \left[ \frac{P_\theta\left(Y \mid \widetilde{\mathcal{G}}\right)}{P(Y)} \right] \\
&= \mathbb{E}_{(Y, \widetilde{\mathcal{G}})} \log \left[ P_\theta\left(Y \mid \widetilde{\mathcal{G}}\right) \right] + H(Y).
\end{aligned}
\tag{26}
$$

As $H(Y)$ remains constant across all data, it can be omitted, resulting in the final formulation of this term:

$$
\mathcal{L}_{pre} := \mathbb{E}_{(Y, \widetilde{\mathcal{G}})} \log \left[ P_\theta\left(Y \mid \widetilde{\mathcal{G}}\right) \right].
\tag{27}
$$

$\square$

### B.2 PROOF OF $\mathcal{L}_{\mathrm{MI}}$

*Proof.* We first use a readout function to obtain the graph representation $z_{\widetilde{G}_{\mathrm{IB}}}$ of the perturbed graph $\widetilde{G}_{\mathrm{IB}}$. And we assume these is no information loss in this process. Therefore we have $I\left(z_{\widetilde{G}_{\mathrm{IB}}}; \widetilde{G}\right) \approx I\left(\widetilde{G}_{\mathrm{IB}}; \widetilde{G}\right)$. Now we bound $I\left(z_{\widetilde{G}_{\mathrm{IB}}}; \widetilde{G}\right)$ using variational approximation:

$$
\begin{aligned}
I\left(z_{\widetilde{G}_{\mathrm{IB}}}; \widetilde{G}\right) &= \iint p\left(z_{\widetilde{G}_{\mathrm{IB}}}, \widetilde{G}\right) \log \frac{p\left(z_{\widetilde{G}_{\mathrm{IB}}} \mid \widetilde{G}\right)}{p\left(z_{\widetilde{G}_{\mathrm{IB}}}\right)} \, dz_{\widetilde{G}_{\mathrm{IB}}} \, d\widetilde{G} \\
&= \iint p\left(z_{\widetilde{G}_{\mathrm{IB}}}, \widetilde{G}\right) \log \frac{p\left(z_{\widetilde{G}_{\mathrm{IB}}} \mid \widetilde{G}\right)}{q\left(z_{\widetilde{G}_{\mathrm{IB}}}\right)} \, dz_{\widetilde{G}_{\mathrm{IB}}} \, d\widetilde{G} \\
&\quad + \iint p\left(z_{\widetilde{G}_{\mathrm{IB}}}, \widetilde{G}\right) \log \frac{q\left(z_{\widetilde{G}_{\mathrm{IB}}}\right)}{p\left(z_{\widetilde{G}_{\mathrm{IB}}}\right)} \, dz_{\widetilde{G}_{\mathrm{IB}}} \, d\widetilde{G} \\
&= \mathbb{E}_{p(\widetilde{G})} \left[ \mathrm{KL} \left( p\left(z_{\widetilde{G}_{\mathrm{IB}}} \mid \widetilde{G}\right) \| q\left(z_{\widetilde{G}_{\mathrm{IB}}}\right) \right) \right] \\
&\quad - \mathbb{E}_{p\left(z_{\widetilde{G}_{\mathrm{IB}}} \mid \widetilde{G}\right)} \left[ \mathrm{KL} \left( p\left(z_{\widetilde{G}_{\mathrm{IB}}}\right) \| q\left(z_{\widetilde{G}_{\mathrm{IB}}}\right) \right) \right] \\
&\leq \mathbb{E}_{p(\widetilde{G})} \left[ \mathrm{KL} \left( p\left(z_{\widetilde{G}_{\mathrm{IB}}} \mid \widetilde{G}\right) \| q\left(z_{\widetilde{G}_{\mathrm{IB}}}\right) \right) \right],
\end{aligned}
\tag{28}
$$

where $q\left(z_{\widetilde{G}_{\mathrm{IB}}}\right)$ is the variational approximation to $p\left(z_{\widetilde{G}_{\mathrm{IB}}}\right)$. And the inequality is due to the fact that Kullback-Leibler divergence is non-negative. We assume that $q\left(z_{\widetilde{G}_{\mathrm{IB}}}\right)$ is a noninformative

distribution following VIB Alemi et al. (2016). That is, we obtain $q\left(z_{\widetilde{G}_{\mathrm{IB}}}\right)$ by aggregating the node representations in a fully perturbed graph. The noise $\epsilon_{\widetilde{G}} \sim \mathcal{N}\left(\mu_h, \sigma_h^2\right)$ is sampled from the Gaussian distribution. $\mu_h, \sigma_h^2$ are mean and variance of $h_j$ in $\widetilde{G}$.

When we choose sum pooling as the readout function, we have:

$$q\left(z_{\widetilde{G}_{\mathrm{IB}}}\right) = \mathcal{N}\left(m_{\widetilde{G}}\mu_h, m_{\widetilde{G}}\sigma_h^2\right). \tag{29}$$

This is because the summation of Gassian distributions is also a Gaussian distribution. Then, for $p\left(z_{\widetilde{G}_{\mathrm{IB}}} \mid \widetilde{G}\right)$, we have:

$$\begin{aligned} &p\left(z_{\widetilde{G}_{\mathrm{IB}}} \mid \widetilde{G}\right) \\ &= \mathcal{N}\left(m_{\widetilde{G}}\mu_h + \sum_{j=1}^{m_{\widetilde{G}}} \lambda_j h_j - \sum_{j=1}^{m_{\widetilde{G}}} \mu_h \lambda_j, \sum_{j=1}^{m_{\widetilde{G}}} (1-\lambda_j)^2 \sigma_h^2\right). \end{aligned} \tag{30}$$

Plug Equation 29 and Equation 30 into Equation 28 and we have:

$$\begin{aligned} &I\left(z_{\widetilde{G}_{\mathrm{IB}}}; \widetilde{G}\right) \\ &\leq \int p(\widetilde{G})\left(-\frac{1}{2}\log A_{\widetilde{G}} + \frac{1}{2m_{\widetilde{G}}}A_{\widetilde{G}} + \frac{1}{2m_{\widetilde{G}}}B_{\widetilde{G}}^2\right) \mathrm{d}\widetilde{G} \\ &\quad + \int \frac{1}{2}p(\widetilde{G})\log m_{\widetilde{G}} \, \mathrm{d}\widetilde{G} \\ &= \int p(\widetilde{G})\left(-\frac{1}{2}\log A_{\widetilde{G}} + \frac{1}{2m_{\widetilde{G}}}A_{\widetilde{G}} + \frac{1}{2m_{\widetilde{G}}}B_{\widetilde{G}}^2\right) \mathrm{d}\widetilde{G} + C, \end{aligned} \tag{31}$$

where $A_{\widetilde{G}} = \sum_{j=1}^{m_{\widetilde{G}}}(1-\lambda_j)^2$ and $B_{\widetilde{G}} = \frac{\sum_{j=1}^{m_{\widetilde{G}}}\lambda_j(h_j - \mu_h)}{\sigma_h}$. $C$ is a constant and can be ignored in the optimization process.

$\square$

## C  THE DETAILED FEATURES FOR ATOMS, BONDS AND MOLECULAR GLOBAL

A comprehensive overview of the selected atom, bond, and global input features is presented in Table 7. The initial step involves the conversion of the SMILES string of both solute and solvent into a graph structure using the RDKit package. This package is employed not only for graph creation but also for the computation of atom and bond features for each graph. The selection of features was restricted to those computable in RDKit to mitigate the computational expenses associated with performing quantum mechanics calculations for the entire dataset. In order to standardize the lengths of the bond, atom, and global feature vectors, a linear transformation is applied to each vector before the commencement of the message-passing steps.

## D  EXPERIMENTAL SETTINGS

In this section, we will provide a comprehensive overview of our experimental setup. Section D.1 will provide detailed information about all the datasets utilized in the experiments. Subsequently, Section D.2 will offer a basic introduction to the baseline methods incorporated in our study. Following that, Section D.3 will delineate the diverse hyperparameters employed in the network architecture of our model. Additionally, it will elucidate the search space for hyperparameters and present the optimal hyperparameters.

Table 7: Atoms (nodes), bonds (edges), and global features for molecular representation

| Atomic features ($\mathcal{V}$) | Bond features ($\mathcal{E}$) | Global features ($\mathcal{U}$) |
|---|---|---|
| Atomic species | Bond type | Total No. of atoms |
| No. of bonds | Conjugated status | Total No. of bonds |
| No. of bonded H atoms | Ring size | Molecular weight |
| Ring status | Stereo-chemistry | – |
| Valence | – | – |
| Aromatic status | – | – |
| Hybridization type | – | – |
| Acceptor status | – | – |
| Donor status | – | – |
| Partial charge | – | – |

## D.1 DATASETS

- **ZhangDDI** Zhang et al. (2017) is a small-scale dataset, including 548 drugs with 48,548 pairwise interaction data points, encompassing various types of similarity information for these drug pairs.

- **ChChMiner** Zitnik et al. a medium-scale dataset, comprises 1,514 drugs and 48,514 labeled DDIs, sourced from drug labels and scientific publications.

- **DeepDDI** Ryu et al. (2018) is a larger-scale dataset with 1,704 drugs and 192,284 labeled DDIs, along with comprehensive side-effect information.

These datasets provide detailed drug information, including SMILES string representations, forming a robust foundation for evaluating the proposed model.

## D.2 BASELINES

In this chapter, we will provide a brief introduction to the baseline models mentioned in the experimental section. In our extensive assessment, our model is compared with eight advanced DDI event prediction methods, all leveraging molecular graphs as input features.

**GoGNN.** Wang et al. (2020a) It extracts features from structured entity graphs and entity interaction graphs in a hierarchical manner. We also propose a dual attention mechanism that enables the model to preserve the importance of neighbors in both levels of the graph.

**DeepDDI.** Ryu et al. (2018)It is based on the structural similarity profile between input drugs and others.

**MHCADDI.** Deac et al. (2019) A gated information transfer neural network is used to control the extraction of substructures and then interact based on an attention mechanism.

**SSI-DDI.** Nyamabo et al. (2021) it use a 4-layer GAT network to extract substructures at different levels, and finally complete the final prediction based on the co-attention mechanism.

**CGIB.** Lee et al. (2023b) Based on the graph conditional information bottleneck theory, conditional subgraphs are extracted to complete the interaction between molecules.

**CMRL.** Lee et al. (2023c) it detects the core substructure that is causally related to chemical reactions. we introduce a novel conditional intervention framework whose intervention is conditioned on the paired molecule. With the conditional intervention framework.

**MDF-SA-DDI.** Lin et al. (2022), achieving DDI prediction by incorporating multi-source drug fusion, multi-source feature fusion and transformer self-attention mechanism.

**DSN-DDI.** Li et al. (2023) it employs local and global representation learning modules iteratively and learns drug substructures from the single drug ('intra-view') and the drug pair ('inter-view') simultaneously.

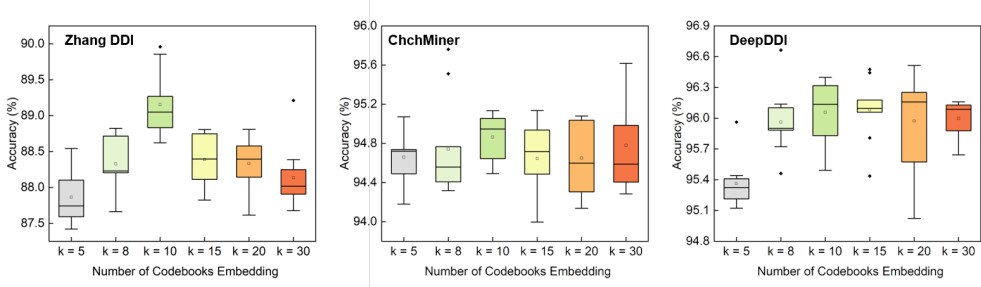

Figure 6: Test results of varying numbers of environmental vectors in the environment codebook in a transductive setting.

### D.3 PARAMETER SETTING

**Model architecture.** For intramolecular message passing, we employ a 3-layer Gated Graph Convolutional Network (GatedConv). For intermolecular message passing, we utilize a 3-layer Graph Attention Network (GAT). As for the pooling layer, we opt for the set2set network. The detatiled hyperparameters are present in Table 8.

**Model Training.** The model is trained using the Adam optimizer Kingma & Ba (2014) with an initial learning rate of $1 \times 10^{-4}$, which is increased to 0.5, employing a batch size of 32. Binary cross-entropy loss (BCE) is utilized as the training loss function. Training is terminated if the validation error does not decrease for 150 epochs or if the maximum training limit of 300 epochs is reached. I2Mole is implemented in the PyTorch framework and executed on Tesla A100 40GB hardware.

Table 8: Hyperparameter specifications.

| Network layer hyperparameters | | | | | | | | |
|---|---|---|---|---|---|---|---|---|
| **GatedConv** | | | **GAT** | | | **FC** | | |
| Num-layers | **3**, 4, 5 | | Num-layers | **3**, 4, 5 | | Num-layers | 3, **4**, 5 | |
| Hidden-size | 200, **400**, 600, 800 | | Hidden-size | 200, **400**, 600, 800 | | Dropout | 0.5 | |
| Layers | **3**, 4, 5 | | Layers | **3**, 4, 5 | | Hidden-size | 300, **400**, 500, 600 | |
| Activation | LeakyReLU | | Activation | LeakyReLU | | | | |
| **Training hyperparameters** | | | | | | | | |
| Batch-size | 32 | | Learning rate | **0.0001**, 0.0005, 0.01, 0.005 | | $\beta$ | 0, $1e^{-5}$,**$1e^{-4}$**, **$1e^{-3}$**, $1e^{-1}$ | |
| **Environment codebook hyperparameters** | | | | | | | | |
| Num of environment | 5, 8, **10**, 15, 20, 30 | | $\gamma$ | $2e^{-5}, 1e^{-4}, 2e^{-4}, 5e^{-4}, 1e^{-3}$ | | | | |

## E RESULTS SIGNIFICANCE ANALYSIS

In the transductive setting, which is a conventional testing method where the dataset is randomly divided into training, validation, and test sets, we repeated the experiment 8 times and calculated the mean, variance, and p-value of the ACC, as shown in the Table 9, compared to the second-best model, I2Mole improved by 1.03% on the large-scale DeepDDI dataset, with an average improvement of approximately 0.05. The p-values are less than 0.05, proving that the improvements of our model are statistically significant. Therefore, although this is a limited improvement, the results are substantial and significant.

In the inductive setting, which is a commonly used generalization testing method, a molecule (type 1) or a molecule pair (type 2) is removed in the test set to verify the model's generalization ability. Notably, in the generalization tests, I2Mole achieved an average improvement of 1.31% in type 1 scenarios and 1.33% in type 2 scenarios. This demonstrates I2Mole's ability to generalize to unseen molecules.

In domain generalization experiments, which is a challenging testing method, we trained and tested on datasets from different domains to verify I2Mole's generalization ability. Clearly, I2Mole achieved the best generalization results, with an average improvement of 4.6% (Acc index), significantly outperforming the improvements in transductive and inductive settings.

Table 9: Significant difference analysis.

|  | Model Performance | | |
|---|---|---|---|
|  | **ZhangDDI** | **ChchMiner** | **DeepDDI** |
| **Second-best model** | $87.78_{(0.37)}$ | $94.63_{(0.21)}$ | $95.76_{(0.72)}$ |
| **Our model** | $88.64_{(0.24)}$ | $95.34_{(0.19)}$ | $96.51_{(0.14)}$ |
| **P-value** | 1.84E-08 | 1.27E-03 | 1.53E-02 |

## F    SENSITIVITY ANALYSIS

We conduct an in-depth investigation into the effect of varying the number of environment embeddings on our model's performance, as depicted in Figure 6. The results demonstrate that altering the number of environment embeddings has a negligible impact on test performance across three different-sized test datasets, underscoring the robustness of our model. The optimal performance is achieved when the number of codebook vectors is set to 10, which we have adopted for our final model configuration.

Table 10: Sensitivity analysis for retained relational edges ratio.

|  | ZhangDDI | | |
|---|---|---|---|
|  | **ACC** | **AUROC** | **F1** |
| Top_s = 5% | $88.52_{(0.08)}$ | $95.22_{(0.02)}$ | $85.75_{(0.06)}$ |
| Top_s = 10% | $88.69_{(0.25)}$ | $95.31_{(0.03)}$ | $85.82_{(0.20)}$ |
| Top_s = 20% | $88.81_{(0.01)}$ | $95.03_{(0.07)}$ | $85.87_{(0.07)}$ |
| Top_s = 30% | $88.54_{(0.12)}$ | $95.69_{(0.12)}$ | $85.53_{(0.13)}$ |
| Top_s = 50% | $88.34_{(0.14)}$ | $95.39_{(0.10)}$ | $85.42_{(0.11)}$ |

Furthermore, we investigated the impact of varying the proportion of retained relational edges during inter-molecular message passing on the model's performance, as illustrated in Table 10. The results indicate that gradually increasing the proportion of retained relational edges enhances the model's performance. However, beyond a certain threshold, further increments lead to a noticeable decline in performance. Consequently, we selected 20% as the optimal parameter for the proportion of retained relational edges.

A crucial parameter in this context is the number of environmental samples, denoted as $\theta$. Increasing the number of sampled environments expands the range of simulated molecular interactions under various conditions, though it also introduces additional training overhead. To identify the optimal value of $\theta$, we systematically evaluated the impact of different sampling quantities, ranging from 2 to 6, within the framework of an environment codebook size of $\mathbf{k} = 10$, as shown in Table 11. Based on the results, we have determined the optimal value of $\theta$ to be 4.

Table 11: Sensitivity analysis for the sampling numbers of environmental embedding.

|  | ZhangDDI | | | ChchMiner | | | DeepDDI | | |
|---|---|---|---|---|---|---|---|---|---|
|  | **ACC** (↑) | **AUROC** (↑) | **F1** (↑) | **ACC** (↑) | **AUROC** (↑) | **F1** (↑) | **ACC** (↑) | **AUROC** (↑) | **F1** (↑) |
| $\theta = 2$ | $88.12_{(0.11)}$ | $94.79_{(0.14)}$ | $85.03_{(0.06)}$ | $94.67_{(0.14)}$ | $98.61_{(0.21)}$ | $95.83_{(0.11)}$ | $96.48_{(0.11)}$ | $97.11_{(0.10)}$ | $98.75_{(0.06)}$ |
| $\theta = 3$ | $88.17_{(0.15)}$ | $94.74_{(0.10)}$ | $85.07_{(0.20)}$ | $94.43_{(0.14)}$ | $98.68_{(0.07)}$ | $95.71_{(0.08)}$ | $96.23_{(0.10)}$ | $96.93_{(0.18)}$ | $98.80_{(0.10)}$ |
| $\theta = 4$ | $88.27_{(0.13)}$ | $94.86_{(0.12)}$ | $85.41_{(0.10)}$ | $94.68_{(0.09)}$ | $98.85_{(0.22)}$ | $95.86_{(0.24)}$ | $96.25_{(0.23)}$ | $96.97_{(0.24)}$ | $98.79_{(0.15)}$ |
| $\theta = 5$ | $88.29_{(0.13)}$ | $94.87_{(0.13)}$ | $85.41_{(0.12)}$ | $94.38_{(0.19)}$ | $98.82_{(0.18)}$ | $95.61_{(0.10)}$ | $96.18_{(0.12)}$ | $96.89_{(0.12)}$ | $98.82_{(0.08)}$ |
| $\theta = 6$ | $88.24_{(0.14)}$ | $94.85_{(0.15)}$ | $85.09_{(0.10)}$ | $94.57_{(0.16)}$ | $98.76_{(0.10)}$ | $95.78_{(0.08)}$ | $96.34_{(0.12)}$ | $97.00_{(0.13)}$ | $98.73_{(0.11)}$ |

## G    VQ MODULE ANALYSIS

To further illustrate the role of the VQ module, we introduced extra two variants for comparison: (1) RD Noise Variant: In this version, noise in the environment codebook is entirely random, mimicking the effects of random noise injection. (2) Instance-Dependent (ID) Noise Variant: Here, we sampled new environments from the environment codebook and added them as small perturbations to the

instance-dependent environment. The Gaussian distribution of this noise is determined by the mean and variance of subgraph node vectors, emphasizing instance-dependent noisy perturbations.

Table 12: Performance comparison across different DDI datasets.

| Method | ZhangDDI Acc (↑) | AUROC (↑) | ChchMiner Acc (↑) | AUROC (↑) | DeepDDI Acc (↑) | AUROC (↑) |
|---|---|---|---|---|---|---|
| RD noise | $87.21_{(0.11)}$ | $93.76_{(0.13)}$ | $93.47_{(0.08)}$ | $97.52_{(0.07)}$ | $92.39_{(0.38)}$ | $97.01_{(0.39)}$ |
| ID noise | $88.02_{(0.06)}$ | $94.47_{(0.08)}$ | $94.27_{(0.12)}$ | $98.54_{(0.06)}$ | $94.56_{(0.10)}$ | $97.42_{(0.31)}$ |
| Ours | $88.64_{(0.24)}$ | $95.12_{(0.12)}$ | $95.34_{(0.19)}$ | $98.84_{(0.10)}$ | $96.51_{(0.14)}$ | $99.04_{(0.22)}$ |

Our experimental results demonstrate the superiority of the VQ module and the proposed optimization strategy, particularly in improving the model's robustness and ability to generalize across diverse chemical environments.

## H    COMPUTATIONAL COMPLEXITY AND DATA SCALABILITY ANALYSIS

We have included the time and space complexity results for our model and various baseline models, along with a comparison of model parameters in Table 13 and Table 15 . This table clearly showcases the results of various model parameters, time and computational complexity. Compared to other baseline models, I2Mole exhibits a significantly larger number of total parameters, leading to substantially higher time consumption and computational complexity than the other baselines.

As the amount of training data increases and the test data decreases, the model's performance exhibits a notable improvement. It is particularly worth mentioning that the model shows significant gains during the initial stages, but further increasing the training data yields only marginal improvements.

Further, we evaluated the model performance under different training data sizes and model parameter conditions in Table 14. We control the model parameters by adjusting the number of message-passing layers, dimensions of embeddings and feedforward NN.

Alongside increasing the model's complexity, there is a clear rise in computational time, accompanied by performance improvements. This could be attributed to the more intricate models being able to capture deeper inter-molecular relationships, thereby enhancing performance. However, when the model's parameters are further increased, its performance starts to degrade. This decline is likely due to overfitting, as the model becomes overly complex, leading to difficulties in convergence during training.

Table 13: Computational complexity analysis on I2Mole.

| ZhangDDI | - | - | - | - | - | - | - | - | - |
|---|---|---|---|---|---|---|---|---|---|
| Model Parameter (M) | 23.4 | 26.8 | 29.5 | 31.7 | 35.4 | 38.4 | 40.3 | 44.5 | 49 |
| Time Consumption (h) | 7.4 | 8.3 | 9.5 | 10.4 | 12.17 | 13.23 | 15.6 | 18.4 | 22.3 |
| Performance (ACC) | 83.61 | 85.94 | 87.43 | 88.27 | 88.64 | 88.64 | 88.35 | 87.91 | 86.87 |

Table 14: Data scalability analysis on I2Mole.

| ZhangDDI | - | - | - | - | - | - | - | - | - |
|---|---|---|---|---|---|---|---|---|---|
| Training Data Ratio (%) | 10 | 20 | 30 | 40 | 50 | 60 | 70 | 80 | 90 |
| Test Data Ratio (%) | 45 | 40 | 35 | 30 | 25 | 20 | 15 | 10 | 5 |
| Performance (ACC) | 53.27 | 65.27 | 72.34 | 80.34 | 88.27 | 88.64 | 89.01 | 91.34 | 92.25 |

## I    LIMITATION ANALYSIS

I2Mole, based on the drug pair merged graph, achieves the extraction of rational subgraphs in molecular interactions and combines the trained environment codebook, significantly enhancing

Table 15: Comparison of ZhangDDI, ChchMiner, and DeepDDI across different models

| Model | Metric | ZhangDDI | ChchMiner | DeepDDI |
|---|---|---|---|---|
| CGIB | ACC | 87.69 | 94.68 | 95.76 |
| | Time (h) | 1.5 | 0.59 | 3.73 |
| | Memory (G) | 5.1 | 3.9 | 7.4 |
| | Total parameters (M) | 11 | 11 | 11 |
| CRML | ACC | 87.78 | 94.43 | 95.99 |
| | Time (h) | 1.3 | 0.47 | 3.24 |
| | Memory (G) | 4 | 3.4 | 6.1 |
| | Total parameters (M) | 10 | 10 | 10 |
| SSI-DDI | ACC | 86.97 | 93.26 | 94.27 |
| | Time (h) | 1.77 | 0.65 | 4.08 |
| | Memory (G) | 3.1 | 2.7 | 4.4 |
| | Total parameters (M) | 13 | 13 | 13 |
| DSN-DDI | ACC | 87.65 | 94.23 | 93.37 |
| | Time (h) | 1.2 | 0.43 | 3.08 |
| | Memory (G) | 2.9 | 3.6 | 4.1 |
| | Total parameters (M) | 0.19 | 0.19 | 0.19 |
| I2Mole | ACC | **88.64** | **95.34** | **96.51** |
| | Time (h) | 13.23 | 4.9 | 33.07 |
| | Memory (G) | 17 | 13.3 | 11.7 |
| | Total parameters (M) | 35.4 | 35.4 | 35.4 |

generalization capabilities in domain generalization experiments. However, considering the rapid advancements in the pharmaceutical field and real-world prescription scenarios, we foresee improvements to the current framework in three key aspects:

- We aim to acquire more comprehensive data on drug interaction processes and analyses between molecules, addressing the limitations of current research. In practice, patients often have multiple comorbidities requiring the concurrent use of various drug categories. Thus, the interaction system of multiple drugs remains a critical research area.

- Constructing relationship edges in pairs, as previously done, is an effective strategy for predicting properties between molecular pairs. However, this approach significantly increases the graph's complexity, especially for large, intricate molecules, due to the substantial rise in degrees of freedom, leading to higher computational resource and time consumption.

- An equally important aspect is that drugs often function only under specific conditions such as temperature and pH levels. Therefore, we anticipate future work to comprehensively consider the impact of external environments on the functionality of drug molecule pairs, thereby refining the model's capabilities.

## J VISUALIZATION ANALYSIS

Acetaminophen, a widely used medication for pain relief (analgesic) and fever reduction (antipyretic), is frequently found in over-the-counter formulations. However, unexpected drug-drug interactions (DDIs) between acetaminophen and compounds such as Fenoterol, Fosphenytoin, and Ethanol can pose significant threats to patient safety, as depicted in Figure (a). Specifically, the aromatic ring of acetaminophen, along with its surrounding functional groups, is capable of interacting with target molecules, particularly at the central carbon atom bonded to the carboxyl group. This key interaction has been effectively captured by the I2Mole model. Additionally, the core subgraphs extracted by I2Mole exhibit good connectivity and consistent distribution across regions, although the associated weights may vary.

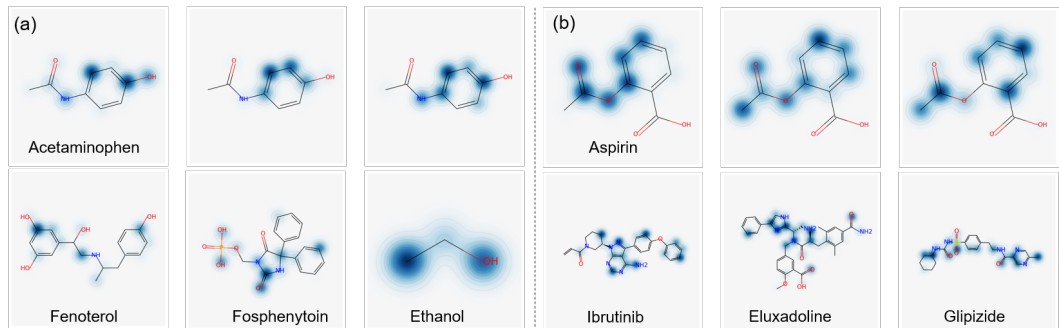

Figure 7: Visualization of the important substructure pairs in six drug pairs. (a) Acetaminophen with Fenoterol, Fosphenytoin, and Ethanol drug ligands. And (b) Aspirin with Ibrutinib, Eluxadoline, and Glipizide drug ligands. The darker the color means the greater the weight.

As demonstrated in Figure (b), In the case of aspirin, a widely used anti-inflammatory drug that also serves as an analgesic, antipyretic, and, at low doses, an antiplatelet agent, its interaction with molecules like Ibrutinib, Eluxadoline, and Glipizide is notably influenced by specific structural features. The aromatic branch of aspirin (excluding the carboxyl group region) is more prone to forming interactions with the nitrogen-containing heterocycles of other molecules. This suggests that, during DDI events, the merged graph substructures of these molecules have an enhanced propensity for direct interaction, leading to increased DDI potential. These observations provide important insights into the structural determinants of DDIs, further emphasizing the predictive capability of the I2Mole model in capturing complex inter-molecular relationships.

