# OpenReview forum: "Capturing substructure interactions by invariant Information Bottle Theory for Generalizable Property Prediction"
_ICLR.cc/2025/Conference — Submitted to ICLR 2025_

### Official Review · Reviewer_DYuZ · 2024-11-01

**Soundness:** 3
**Presentation:** 2
**Contribution:** 4
**Rating:** 5
**Confidence:** 4

**Summary:**

This paper proposes a new method, I2Mole (Interaction-aware Invariant Molecular learning), for predicting molecular interactions, such as drug-drug interactions. This method is based on two main ideas: (1) identifying subgraph pairs (core substructures) that contribute most to prediction accuracy, based on Graph Information Bottleneck (GIB) theory (Yu et al, 2020; 2022b), and (2) implementing out-of-distribution generalization through invariant learning (Wu et al, 2022a) to mitigate environmental factors that may disrupt estimation.

In (1), intramolecular message passing is first conducted for each molecule of a molecular pair, followed by intermolecular message passing, defining a merged graph from non-zero coefficients of intermolecular connections. Core substructure identification is then performed on this graph applying GIB, in particular, "compression via noise injection (Yu et al., 2022b)." In (2), several latent environmental factors in the training data are represented as learnable embedding vectors, and prediction robustness is enhanced by clustering non-core substructure variations through learning vector quantization (LVQ). The method demonstrates superior predictive accuracy over established approaches in drug-drug interaction (DDI) prediction benchmarks.

**Strengths:**

- In molecular interactions, explicitly considering subgraph-level interactions is crucial for both predictive accuracy and interpretability. This study leverages GIB (Yu et al., 2022b) and invariant learning (Wu et al., 2022a) as highly relevant prior knowledge to achieve more precise molecular interaction modeling and predictions that are robust to environmental factors.
- In the presentation of the paper, the motivations and intentions are clearly introduced through examples, followed by technical details, making it easier to grasp the design principles despite the complex methodology.
- Compared to many existing methods, this approach demonstrates outstanding results in DDI prediction benchmarks.

**Weaknesses:**

- Overall, the paper is intriguing, but the description in Section 3.3, one of the two main ideas, is highly unclear and falls short of publication standards. For example, at the start of Section 3.3, it states that \tilde{G}_{IB} is obtained from Equation (19), but in its definition, it is derived via optimization in Equation (12), which is contradictory. Additionally, mutual information is represented as I(Y; \tilde{G}_{IB}) in Equation (12) but as MI(\tilde{G}_{IB}, Y) in Equation (19), showing inconsistency in notation. In Section 3.1, \\tilde{G}_{IB} is defined as a graph, yet in Equations (20)–(22), it is treated as a multivariate vector apparently, with no explanation of this difference within the paper. Furthermore, the description of q(\tilde{G}_{IB}) in Equation (20) as “the distribution of data under environment \tilde{G}_{IB}” is too vague to understand, and defining \tilde{G}_{IB} as a subgraph of \tilde{G} while expressing (\tilde{G}_{IB}, Y) \sim q(\tilde{G}_{IB}) is unclear.

- Numerous simple typos are found throughout the draft. For instance, in the title “Bottle” should be “Bottleneck”; in Section 2.1, “a molecular” should be “a molecule”; “is is” on line 187 should be “is”; parentheses in Equation (6) do not match; “Equation equation” on line 219 should be “equation”; in Equation (19), MI(\tilde{G}_{IB}, Y) should be I(Y; \tilde{G}_{IB}); there is inconsistency on whether \tilde{G}_{IB} is a graph or a vector; and “vertor” on line 479 should be “vector.”

- Regarding Sections 3.2.1 and 3.2.2 on GIB applications, the approach primarily applies Yu et al. (2022b)’s method to a merged graph, which lacks technical originality. It would be better to clarify this point, as the sections present ideas from Yu et al. (2022b) as if they were original derivations, which is misleading. If these points are self-evident, emphasizing the proofs is unnecessary.

- The term “out-of-distribution generalization” requires discussion. Since this paper primarily addresses adaptations to environmental changes within the training data by modeling environment patterns through clustering, it focuses on robustness within the training data rather than an out-of-distribution approach. Typically, “out-of-distribution” refers to data outside the training distribution, so calling this an OOD method could be quite misleading. The original idea in Section 2.3 would be assuming that the external environments can be represented by latent factors, therefore the adjustment of Eq(2) would make sense to achieve unbiased predictions to mitigate any potential OOD biases.

**Questions:**

- What does the first sentence in Section 3.3 mean? Since $\tilde{G}_{GIB}$ is obtained through the optimization in Equation (12), shouldn’t it be referring to that equation rather than Equation (19)? Also, why is the notation for mutual information inconsistent? Any special meaning here?

- In Section 3.3, it seems that \tilde{G}_{GIB} is treated as a vector. How is this derived? If it’s obtained through some form of global pooling, the procedure should be clearly specified and reflected in the notation. The paper defines \tilde{G}_{GIB} as a graph, so Equations (20) and (21)–(22) doesn't make any sense.

- The probability distribution $q$ assumed in $(\tilde{G}, Y) \sim q(\tilde{G}_{GIB})$ and the sampling process from it are unclear. With respect to what variable is the expectation $\mathbb{E}$ taken?

- Also, $\tilde{G}_{GIB}$ is a subgraph of a graph from the training data, so it seems it would not contain information beyond the training set. Is it appropriate to refer to this as “out-of-distribution generalization”? If there’s any additional information on this point, please clarify.

---

> ### Author Response · Authors · 2024-11-21
> **Response to **Reviewer DYuZ** (Part I)**
>
> Thank you for the valuable feedback. I will address each of your concerns in detail.
>
> > ***W1&W2&Q1**: Equation (19) needs revision, and mutual information symbols should be consistent*
>
> Thank you for your suggestion. In fact, Equation 19 corresponds to the first term in **Equation 12**, which represents the maximization of mutual information between the extracted core subgraph and the label $Y$, thereby ensuring that $\mathcal{G} _ {\mathrm{sub}}$ retains the most relevant information for $Y$. The second term in Equation 12 is the compression term, which aims to minimize the structure of $\mathcal{G} _ {\mathrm{sub}}$, retaining only the most essential parts, $\mathcal{G} _ {\mathrm{IB}}$.
>
> We have revised the corresponding section to improve the readability and coherence of the manuscript. Additionally, the mutual information symbols have been standardized in the revised version for consistency.
>
> > ***W1&W2&Q2**: $\tilde{G}_{IB}$ is defined as a graph, yet in Equations (20)– (22), it is treated as a multivariate vector apparently. with no explanation of this difference within the paper*
>
> Thank you for your professional suggestion. We have clarified the distinction between graph representation and vector representation in revised version: *“Additionally, the set2set network is utilized to pool $\widetilde{\mathcal{G}} _ \text{IB}$, $\widetilde{\mathcal{G}} _ \text{env}$, and $\widetilde{\mathcal{G}}$, resulting in the substructure representation vectors $s _ {\text{IB}}$, $s_{\text{env}}$, and $s _ {\mathcal{G}}$.”* Furthermore, the related formula symbols in the revised manuscript have been updated for improved rigor and consistency throughout the paper.
>
> > ***W1&Q3**: Furthermore, the description of $q(\tilde{G} _ {IB})$ in Equation (20) as “the distribution of data under environment $\tilde{G} _ {IB}$” is too vague to understand. With respect to what variable is the expectation E taken?*
>
> In our method, we assume that the merged molecular graph $\widetilde{\mathcal{G}}$ can be decomposed into two parts: the rationale and the environment. The non-core part is considered the latent environment $\widetilde{\mathcal{G}} _ \text{env}$. The $\widetilde{\mathcal{G}} _ \text{env}$ can be combined with various rationales to form new molecules representation, collectively constituting the distribution $q(\widetilde{\mathcal{G}} _ \text{env})$.
>
> Directly computing the expectation $\mathbb{E}$ by traversing all molecules in the distribution $q(\widetilde{\mathcal{G}} _ \text{env})$ is challenging due to the infinite nature of the space. To address this issue, we transform the problem into traversal sampling within a finite space. By doing so, we map the infinite potential environments into a fixed number of $M$ environments, enabling the expectation to be solvable according to **Equation 23**.
>
>
> >***W1&W2**: defining $\tilde{G} _ {IB}$ as a subgraph of $\tilde{G}$ while expressing $(\tilde{G}{IB}, Y) \sim q(\tilde{G}_{IB})$ is unclear.*
>
> We sincerely apologize for the unclear expression that may have caused confusion. The notation $(\widetilde{\mathcal{G}}, \mathbf{Y}) \sim q(\widetilde{\mathcal{G}} _ \text{env})$ indicates that $\widetilde{\mathcal{G}}$ is sampled from the space defined by $q(\widetilde{\mathcal{G}} _ \text{env})$.
>
> > ***W1&Q3**: Furthermore, the description of $q(\tilde{G}{IB})$ in Equation (20) as “the distribution of data under environment $\tilde{G} _ {IB}$” is too vague to understand. With respect to what variable is the expectation E taken?*
>
> In our method, we assume that the merged molecular graph $\widetilde{\mathcal{G}}$ can be decomposed into two parts: the rationale and the environment. The non-core part is considered the latent environment $\widetilde{\mathcal{G}} _\text{env}$. The $\widetilde{\mathcal{G}} _ \text{env}$ can be combined with various rationales to form new molecules representation, collectively constituting the distribution $q(\widetilde{\mathcal{G}} _\text{env})$.
>
> Directly computing the expectation $\mathbb{E}$ by traversing all molecules in the distribution $q(\widetilde{\mathcal{G}} _ \text{env})$ is challenging due to the infinite nature of the space. To address this issue, we transform the problem into traversal sampling within a finite space. By doing so, we map the infinite potential environments into a fixed number of $M$ environments, enabling the expectation to be solvable according to **Equation 23**.
>
> > ***W1&W2**: defining $\tilde{G}{IB}$ as a subgraph of $\tilde{G}$ while expressing $(\tilde{G}{IB}, Y) \sim q(\tilde{G}_{IB})$ is unclear.*
>
> We sincerely apologize for the unclear expression that may have caused confusion. The notation $(\widetilde{\mathcal{G}}, \mathbf{Y}) \sim q(\widetilde{\mathcal{G}} _ \text{env})$ indicates that $\widetilde{\mathcal{G}}$ is sampled from the space defined by $q(\widetilde{\mathcal{G}} _ \text{env})$.

---

> > ### Author Response · Authors · 2024-11-21
> > **Response to **Reviewer DYuZ** (Part II)**
> >
> > > ***W2**: Numerous simple typos are found throughout the draft.*
> >
> > - In the title “Bottle” should be “Bottleneck”;
> >
> > Thank you for your suggestion. The title has been updated, replacing “Bottle” with “Bottleneck” for accuracy.
> >
> > - in Section 2.1, “a molecular” should be “a molecule”;
> >
> > Thank you for the clarification. The term has been updated to "a molecule" in the revised version.
> >
> > - “is is” on line 187 should be “is”;
> >
> > We have removed the redundant "is" on line 187. Thank you for pointing this out.
> >
> > - parentheses in Equation (6) do not match;
> >
> > We have revised the equation to:
> >
> > $$
> > e_{ij}^{\prime} = e_{ij} + {\rm LeakyReLU}[{\rm FC}(v_i + v_j) + [{\rm FC}(e_{ij}) + [{\rm FC}(u)]],
> > $$
> >
> > and ensured that the square brackets are properly aligned in the revised version. Thank you for pointing this out.
> >
> > - “Equation equation” on line 219 should be “equation”;
> >
> > We have removed the redundant "Equation" on line 219.
> >
> > - And “vertor” on line 479 should be “vector.”
> >
> > The word "vertor" on line 479 has been corrected to "vector."
> >
> > Additionally, in the revised version, we conducted a thorough review of the entire manuscript and have corrected all grammatical and spelling errors. Thank you for your careful feedback!
> >
> > > ***W3**: Distinguishing Yu et al. (2022b)’s work in Sections 3.2.1 and 3.2.2 on GIB applications.*
> >
> > **In Section 3.2.1**, we reference Yang et al. (2022b), which proposed two key assumptions for molecular OOD problems: the **Sufficiency assumption** and the **Invariance assumption**. In this section, we focus on optimizing Equation (13) to ensure the **Sufficiency assumption** is satisfied. However, our method for extracting core substructures differs significantly from Yang et al.’s approach, which represents one of the main innovations of our work. Unlike their method, our core substructure extraction strategy introduces distinct improvements tailored for the problem at hand.
> >
> > **In Section 3.2.2**, we reference Yu et al. (2022b), an important milestone in this domain, which introduced noise injection to optimize mutual information. Specifically, VGIB employs noise injection to modulate the information flow from the input graph to the perturbed graph. However, their approach relies on Gaussian distributions over node features for noise injection, which lacks explicit modeling of the actual chemical space. Building upon their work, we integrate edge information into the computation of node importance (**Equations 14 & 15**), which is particularly critical for molecular relational learning. This integration accounts for intermolecular interactions that influence the selection of core substructures, marking a significant difference from their method.
> >
> > Meanwhile, thank you for your suggestion. We have decided not to emphasize the proof part here in the revised version.
> >
> > > ***W4&Q4**: The term “out-of-distribution generalization” maybe not appropriate.*
> >
> > Thank you for your suggestion. Our work aligns more closely with **Stable Learning** within the OOD domain, making the term "Out-Of-Distribution" less appropriate in certain contexts. To ensure a more precise and rigorous description, we have carefully revised the usage of "Out-Of-Distribution" in the updated manuscript, replacing or modifying it where necessary.
> >
> > ---------------
> >
> > We greatly appreciate your insightful and helpful comments, as they will undoubtedly help us improve the quality of our article. If our response has successfully addressed your concerns and clarified any ambiguities, we respectfully hope that you consider raising the score. Should you have any further questions or require additional clarification, we would be delighted to engage in further discussion. Once again, we sincerely appreciate your time and effort in reviewing our manuscript. Your feedback has been invaluable in improving our research.

---

> ### Comment · Reviewer_DYuZ · 2024-11-22
>
> Thank you for your response! I feel I now have a better understanding of the paper’s content. However, there’s one big issue related to the main claim of the paper that remains unclear, so I’d like to ask for further clarification.
>
> As indicated by its inclusion in the Keywords section, I believe the concept of Out-of-Distribution (OOD) is one of the key focuses of this paper. In the main text, the authors state that “the OOD problem can be formally defined as follows,” which resulted in the main problem defined by Eq (20). I guess that the solvable approximation of (20) would be Eq (25).
>
> Regarding this main formulation, **the meaning and assumptions behind the term "environment" in Eq (20) still remain unclear and not well defined to me, as pointed out by most other reviewers**.
>
> - What exactly does $\tilde{G}_{env}$ in the main problem of Eq. (20) mean? A graph or a subgraph? A vector? A set of graphs or vectors? Or something else?
> - Is the support E an infinite set?
> - Or, is Eq (20) just an abstract metaphor or something?
> - When the minimization of Eq (20) is approximated by Eq (23), the paper refers to "environmental samples." Does "out-of-distribution" (or environments) in this paper simply refer to **the internal variations within the training distribution**, and is it correct to interpret the assumption of this paper as **these variations being well-represented by M codes in the code book**? (I still felt that this was not "out of" the training distribution at all.)

---

> ### Author Response · Authors · 2024-11-22
> **Response to **Reviewer DYuZ** (Part III)**
>
> Thank you very much for your response. We are glad to hear that our reply has addressed most of your concerns. Regarding the remaining issue, we are more than happy to provide further clarification to facilitate your understanding.
>
> ***1.** As indicated by its inclusion in the Keywords section, I believe the concept of Out-of-Distribution (OOD) is one of the key focuses of this paper.*
>
> We have accepted your suggestion. In the revised version, we have removed references related to OOD words, including  **Keywords section**. The revised version is still being finalized, and we will upload it promptly once completed.
>
> ***2.** In the main text, the authors state that “the OOD problem can be formally defined as follows,” which resulted in the main problem defined by Eq (20). I guess that the solvable approximation of (20) would be Eq (25).*
>
> In the original version, Equation (20) is approximately solved by Equation (24), i.e., $\mathcal{L} _ {inv}$. This equation represents placing the extracted $\mathcal{G} _ {\mathrm{IB}}$ into different environments $env _ {i}$ learned through VQ modules to obtain robust representations.
>
>
> Equation (25) defines the overall loss $\mathcal{L} _ {total}$, which includes $\mathcal{L} _ {inv}$, along with $\mathcal{L} _ {pre}$, $\mathcal{L} _ {MI}$, and $\mathcal{L} _ {vq}$:
>
> - $\mathcal{L} _ {pre}$ can be modeled as the cross-entropy loss for classification tasks and the mean squared error loss for regression tasks.
> - $\mathcal{L} _ {MI}$ represents the KL divergence between the extracted substructures and the remaining subgraph, encouraging substructure compression.
> - $\mathcal{L} _ {vq}$ aims to optimize the environment vectors in the codebook by mapping multiple environments from the training set into a finite set of discrete environment codes.
>
> ***3.** What exactly does $\tilde{G}_{env}$ in the main problem of Eq. (20) mean? A graph or a subgraph? A vector? A set of graphs or vectors? Or something else? is Eq (20) just an abstract metaphor or something?*
>
> In Equation (20), we assume that the merged graph can be decomposed into substructures $\mathcal{G} _ {\mathrm{IB}}$ and $\mathcal{G} _ {\mathrm{env}}$, where $\mathcal{G} _ {\mathrm{IB}}$ represents the core substructure and $\mathcal{G} _ {\mathrm{env}}$ represents the non-core part (*i.e.*, $\mathbf{h}^{r} _ {i}$ in Equation (16)). In the subsequent implementation, we pool $\mathcal{G} _ {\mathrm{env}}$  into vectors $\mathcal{s} _ {\mathrm{env}}$ (**In revised version**). Equation (20) could be considered as an abstract concept that conveys our intention to traverse all environments, aiming to minimize prediction error (risk function) even in the worst-case scenario. Here, "environment" is an abstract concept, and in our approach, we consider the non-core substructure $\mathcal{G} _ {\mathrm{env}}$ as the environment source.
>
> Specifically, the environment vector $\mathcal{s} _ {\mathrm{env}}$ is obtained by applying a **pooling operation** to $\mathcal{G} _ {\mathrm{env}}$. Since there are numerous $\mathcal{G} _ {\mathrm{env}}$ structures in the training data, this also generates many $\mathcal{s} _ {\mathrm{env}}$ vectors. To manage this complexity, we use a VQ (Vector Quantization) module to map them to a finite set of $M$ embedding vectors in the codebook $W$.
>
> In the subsequent sections, specifically Equations (21-23), these operations should be treated as vector operations rather than $\mathcal{G} _ {\mathrm{env}}$ and $\mathcal{G} _ {\mathrm{IB}}$. Therefore, we have corrected the equations in the revised version to reflect this more rigorously.
>
> To enhance the clarity of **Section 3.3**, we have reorganized and revised the content to improve readability and prevent misunderstandings.
>
> ***4.** Is the support E an infinite set?*
>
> $\mathbf{E}$ depends on the number of samples in the training set. Theoretically, it represents an infinite potential chemical space; however, in practice, it is equal to the number of training samples.

---

> > ### Author Response · Authors · 2024-11-22
> > **Response to **Reviewer DYuZ** (Part IV)**
> >
> > ***5.** When the minimization of Eq (20) is approximated by Eq (23), the paper refers to "environmental samples." Does "out-of-distribution" (or environments) in this paper simply refer to the internal variations within the training distribution, and is it correct to interpret the assumption of this paper as these variations being well-represented by M codes in the code book? (I still felt that this was not "out of" the training distribution at all.)*
> >
> > We concatenated the core substructures from the training set, $\mathcal{G} _ {\mathrm{IB}}$ (the abstract concept, actually this is a vector $\mathcal{s} _ {\mathrm{IB}}$), with different environments, $\mathcal{G} _ {\mathrm{env}}$ (the abstract concept, actually this is a vector $\mathcal{s} _ {\mathrm{env}}$). This can be understood as the model learning new combined molecules that are beyond the original training set, effectively expanding the training distribution, which we initially referred to as "out of" the training distribution. However, we also acknowledge your suggestion that our work may align more closely with **Stable Learning** within the OOD domain, making the term "Out-Of-Distribution" less appropriate in certain contexts. To ensure a more precise and rigorous description, we have carefully revised the use of "Out-Of-Distribution" in the updated manuscript, replacing or modifying it where necessary.
> >
> > -----
> > Lastly, we would like to mention that our revised version has undergone comprehensive modifications and improvements. It will likely be uploaded in a few hours to provide you with a clearer understanding of the changes and corrections we have made.
> >
> > ---
> > Thank you very much for your feedback and response. Your comments are invaluable and essential for improving the quality of the manuscript. If you have any further concerns, please do not hesitate to contact me at your earliest convenience. Hope you a nice day and thanks again!

---

> ### Comment · Reviewer_DYuZ · 2024-11-25
>
> Thank you for the clarification. I understand the explanation about modeling variations in the environment outside the core part using VQ in the training data to stabilize predictions. However, if that’s the case, Observation 7 seems like an expected outcome. This paper appears to focus more on technically implementing robustness, and the discussion on how much of the chemical mechanisms of interaction—the original motivation mentioned in the Introduction and Figure 1—has been captured feels insufficient. Furthermore, removing “OOD prediction” arguments, which was one of the main motivations at the time of submission, and revising the manuscript accordingly seems like a major revision that would typically require re-review.

---

> > ### Author Response · Authors · 2024-11-25
> > **Response to **Reviewer DYuZ** (Part V)**
> >
> > Thank you very much for your insightful feedback. We are very pleased to hear that the previous concerns regarding the methodology have been resolved. As for the new concern you raised, we fully understand and are more than willing to provide further clarification and make necessary revisions to improve the quality of the manuscript.
> >
> > ***1.** Observation 7 seems like an expected outcome.*
> >
> > We can understand your concern that Observation 7 may seem like an expected outcome under this framework. We sincerely apologize for our lack of depth in expressing the content of Observation 7, which may have led to this misunderstanding. Here, I will further clarify our intent and analysis process.
> >
> > As you have noted, we constructed an environment codebook$W$ to obtain environment vectors from molecular pairs and clustered them into $M$ environment vectors (in our case, $M$=10). This process led to Figure 5(a). Naturally, since the environment vectors are derived from non-core substructures $\mathcal{G}_{\mathrm{env}}$, the statement "_Environmental coding is consistent with real chemical scenarios_" might seem self-evident, as you mentioned. This may have contributed to the perception that Observation 7 is an expected outcome under this framework.
> >
> > However, the learning process of the environment codebook is governed by **Equation (21)**, i.e., $\mathcal{L} _ {inv}$. Here, $sg$ represents the stop gradient operator, which acts as the identity during forward computation and has zero partial derivatives, effectively constraining its operand to be a non-updated constant. The first term uses the $l _ 2$ error to move the embedding environment $env _ m$ in the codebook towards $\mathcal{s} _ {\mathrm{env}}$, aiming to update the codebook by mapping the $\mathcal{s} _ {\mathrm{env}}$ values from the training set to the nearest envmenv_m in the codebook. The second term represents a commitment loss, which encourages $\mathcal{s} _ {\mathrm{env}}$ to stay close to the chosen codebook embedding envmenv_m, thereby accelerating the convergence of the codebook. The parameter $\delta$  is a hyperparameter that balances these two terms. It is under the control of $\mathcal{L} _ {inv}$ that we ultimately obtain the environment codebook.
> >
> > To provide an intuitive understanding, we visualize in Figure 5(a) the embeddings of our environment vectors, along with the non-core substructure embeddings $\mathcal{s} _ {\mathrm{env}}$ that are mapped to each environment embedding $env_m$. According to Figure 5(a), the $\mathcal{s} _ {\mathrm{env}}$ values are tightly clustered around their respective environment codes, with clear boundaries between different environment codes and their structure embeddings. This indicates that the codebook updating process is effectively equivalent to clustering the structural embeddings, and the environment codes can be viewed as clustering centers. Thus, the conclusion regarding clustering centers is derived from the analysis of Figure 5(a). Furthermore, we present the distribution of element types corresponding to each environment code in Figure 5(b). We observe that different environment codes tend to encode the local environments of different molecular pairs, as reflected directly in the varying proportions of different elemental types.
> >
> > Based on the above analysis, we arrived at the conclusion in Observation 7. However, we realized that our explanation was too superficial and lacked these critical analytical details in the original version. In the revised manuscript, we will revise this section to include the above analysis, thereby providing more depth and thoughtful insight into Observation 7.
> >
> >
> > ***2.** The original motivation mentioned in the Introduction and Figure 1—feels insufficient.*
> >
> > Regarding the discussion and analysis of molecular interactions, our focus is on emphasizing the impact of inter-molecular interactions, as highlighted in the manuscript's mention of the insufficiency in molecular interaction modeling (**Introduction**). In our work, we adopt a merged graph approach to model molecular interactions, employing TGAT to retain a portion of the relation edges $\mathcal{R}$.
> >
> > When the proportion of $\mathcal{R}$ is too low, it negatively impacts the interaction between molecules, thereby weakening the inter-molecular message passing process. Conversely, when the proportion of $\mathcal{R}$ is too high, individual atoms may become overwhelmed by excessive redundant information from other atoms, which diminishes their intrinsic information. Moreover, this also inevitably leads to a significant increase in model complexity.
> >
> > The experimental results and analysis related to this aspect are provided in **Appendix F**, where we control the message-passing process between molecular pairs by setting different weight thresholds, due to page limitations. Additionally, we have added extra case analyses in **Appendix J** in the revised version to further illustrate how the model extracts subgraphs.

---

> > > ### Author Response · Authors · 2024-11-25
> > > **Response to **Reviewer DYuZ** (Part VI)**
> > >
> > > ***3.**  Removing “OOD prediction” arguments seems like a major revision.
> > >
> > > Thank you for raising this point. I believe you are referring to "OOD generalization," as the term "OOD prediction" does not appear in either the original or the revised version of the manuscript. I would like to take this opportunity to clarify this aspect to avoid any potential misunderstanding.
> > >
> > > Our work has consistently focused on enhancing model generalization, which remains one of our key motivations. This motivation has not changed throughout the revisions. Specifically, we adopted the invariant learning principle to mitigate molecular environmental factors that might disrupt performance. However, since our environment vectors are trained on the training set, it might have led to your perception that our approach is not addressing a typical OOD scenario. Therefore, the core difference seems to lie in whether this problem should be classified as an "OOD scenario" or an "OOD problem."
> > >
> > > In this work, the environment vectors are concatenated with different molecular rationales. By learning these environment vectors and placing $\mathcal{G} _ {\mathrm{IB}}$ in different environments, I2Mole achieve stable and robust outputs. This can be intuitively understood as generating features for new molecular pairs (abstraction conception; as the combination of $\mathcal{G} _ {\mathrm{IB}}$ and $\mathcal{G} _ {\mathrm{env}}$), thereby significantly expanding the training dataset. Such an approach serves as a form of data augmentation, enhancing robustness and stability to mitigate OOD issues [1,2].
> > >
> > > Meanwhile, our main experimental settings, such as molecular scaffold and size splits [3,6], domain shift tests [4], and inductive settings [5], are all commonly used approaches to evaluate OOD generalization. The OOD problem typically refers to the challenge faced when the model encounters samples during testing that do not belong to the training data distribution, which could lead to inaccurate, unreliable, or unstable predictions. Such situations may result in misleading outcomes on unseen data, as the model lacks the ability to generalize adequately to these novel data points. Consequently, we used the term "OOD generalization" in the original manuscript.
> > >
> > > To ensure a more rigorous expression, we have made adjustments in some parts of the text. After reviewing the literature [7,8], we found that **Stable Learning** aligns more closely with our work, as it is a **specific sub-area** within the OOD domain. To ensure a more precise and rigorous description, we have carefully revised the use of "Out-Of-Distribution" in the updated manuscript, replacing or modifying it where necessary. Our goal remains to enhance the model's performance on unseen molecular pairs, given the diverse and complex nature of molecular species in real-world scenarios, which is consistent with the objectives of OOD. However, based on your professional feedback during the rebuttal stage, we acknowledge that the use of "OOD generalization" in some parts of the manuscript was not sufficiently rigorous. Consequently, we have replaced "OOD generalization" with "stable" or "robust" in four locations: the title, the Introduction, Section 3.3, and Observation 6. The overall motivation and experimental aspects of the manuscript remain unchanged.
> > >
> > >
> > >
> > > [1] Data augmentation can improve robustness. NIPS, 2021.
> > >
> > > [2] Investigating the effectiveness of data augmentation from similarity and diversity: An empirical study. Pattern Recognition, 2024.
> > >
> > > [3] MoleOOD: Learning Substructure Invariance for Out-of-Distribution Molecular. NIPS, 2024.
> > >
> > > [4] DSIL-DDI: A Domain-Invariant Substructure Interaction Learning for Generalizable Drug-Drug Interaction Prediction. TNNLS, 2024.
> > >
> > > [5] DSN-DDI: an accurate and generalized framework for drug–drug interaction prediction by dual-view representation learning. Briefings in Bioinformatics, 2023
> > >
> > > [6] DrugOOD: Out-of-Distribution (OOD) Dataset Curator and Benchmark for AI-aided Drug Discovery. AAAI, 2023
> > >
> > > [7] Deep Stable Learning for Out-Of-Distribution Generalization. 2021, CVPR.
> > >
> > > [8] Stable learning establishes some common ground between causal inference and machine learning. 2022, Nature Machine Intelligence.
> > >
> > > ------------------------
> > > We hope that the above response addresses your latest concerns. If you have any questions or need further clarification, please feel free to provide additional guidance or points that need to be addressed. We sincerely appreciate your valuable feedback, which is crucial for improving the quality of the manuscript. Wishing you a wonderful day, and thank you once again!

---

> > > > ### Author Response · Authors · 2024-11-26
> > > > **Response to **Reviewer DYuZ** (Part VII)**
> > > >
> > > > Dear Reviewer,
> > > >
> > > > We have further revised the manuscript and uploaded it to the system for your review. In this revision, we have reorganized and refined the content of Obs.7. Additionally, we emphasized "model generalization" instead of focusing on "OOD generalization" to prevent potential misunderstandings and make it easier for readers to understand and accept our work.
> > > >
> > > > We sincerely thank you once again for your valuable feedback, which has significantly contributed to improving the quality of this manuscript. If you have any further questions or need clarification, please feel free to provide additional guidance or points that need to be addressed. Wishing you a wonderful day, and thank you once again!
> > > >
> > > > Best regards,
> > > >
> > > > The Authors

---

> > > > > ### Author Response · Authors · 2024-11-28
> > > > > **Have our responses adequately addressed your concerns?**
> > > > >
> > > > > **Dear  Reviewer DYuZ**,
> > > > >
> > > > > We have provided detailed responses to your reviews and are pleased to know that our previous replies have addressed many of your concerns. Regarding the new comments on the writing aspects, we would like to know if our responses have adequately addressed your concerns. If you have any further questions or concerns, we are more than happy to continue the discussion.
> > > > >
> > > > > Thank you again for your time and effort in reviewing our manuscript. Your feedback has been instrumental in improving our research!
> > > > >
> > > > > Best regards,
> > > > > Authors

---

> > > > > > ### Comment · Reviewer_DYuZ · 2024-11-28
> > > > > >
> > > > > > Sorry for not responding. I thought the discussion phase had already ended. I acknowledge that I have read all the responses.
> > > > > >
> > > > > > I think I understand what was written in the response. If we decide to use the core structure as the key, handling variations in the non-core structures that disrupt learning stability with learning vector quantization (LVQ), which can localize the representation learning, feels technically incremental to me. I think this would be solid research but incremental, so I’ll leave my score as it is.

---

> > > > > > > ### Author Response · Authors · 2024-12-01
> > > > > > > **Response to **Reviewer DYuZ** (Part Ⅷ)**
> > > > > > >
> > > > > > > **1.** *learning vector quantization (LVQ) to handle the non-core structures could disrupt learning stability.*
> > > > > > >
> > > > > > > Thank you very much for your feedback. We have carefully considered your concern regarding the potential disruption of learning stability when using vector quantization (VQ) to handle the non-core structures. I would like to provide further clarification on this matter:
> > > > > > >
> > > > > > > First, Variational Autoencoders (VAEs) [1] are powerful tools for generative modeling and learning data-efficient representations, with applications in various fields such as image generation, anomaly detection, and data compression. The Vector Quantized VAE (VQ-VAE) [2] is a successful discrete VAE model that uses vector quantization to discretize continuous latent variables, achieving high compression rates while generating compact and meaningful codebooks. It has been widely applied in domains such as computer vision [2], protein-protein interaction prediction [3], and molecular encoding [4]. VQ-VAE aims to learn a progressive, sequential structure for data representation that maximizes the mutual information between the latent representations and the original data, all while maintaining a limited description length.
> > > > > > >
> > > > > > > Given the vast chemical space and the diversity of molecular combinations, we seek to make the latent codes compact, meaningful, and represented by as few bits as possible. At the same time, it is essential for the latent codes to convey as much information as possible, requiring the model to be confident in the latent codes derived from the input. Therefore, we must balance the trade-off between description length and information content in the latent code.
> > > > > > >
> > > > > > > Regarding this issue, VAEs may lose encoding information in certain scenarios, particularly during the optimization process, where the structuring of the latent space could result in information loss. This primarily depends on the trade-off controlled by the **KL divergence**, which encourages the latent variable distribution to approximate a standard normal distribution. This regularization can sometimes constrain the model’s ability to capture intricate details in the data. As a result, the latent space of a VAE may not fully preserve all the information in the input data, especially when the data exhibits complex structures. **However, variants of VAEs, such as the vector quantization approach used in this paper (only encoder), improve the structuring of the latent space by adjusting the weight of the KL divergence and by introducing quantization techniques as in this work. These modifications help avoid oversimplification of the latent variables and contribute to improved model stability [3,4,5]**.
> > > > > > >
> > > > > > > Thus, we believe the concern you raised can be mitigated.
> > > > > > >
> > > > > > >
> > > > > > >
> > > > > > > [1]  Fixing a Broken ELBO. ICML, 2018.
> > > > > > >
> > > > > > > [2] Neural discrete representation learning. NIPS, 2017.
> > > > > > >
> > > > > > > [2] PQ-VAE: Learning Hierarchical Discrete Representations with Progressive Quantization. CVPR, 2024
> > > > > > >
> > > > > > > [3] MAPE-PPI.  MAPE-PPI: Towards Effective and Efficient Protein-Protein Interaction Prediction via Microenvironment-Aware Protein Embedding. ICLR, 2024.
> > > > > > >
> > > > > > > [4] Learning Invariant Molecular Representation in Latent Discrete Space. NIPS, 2024.
> > > > > > >
> > > > > > > [5] Mole-BERT: Rethinking Pre-training Graph Neural Networks for Molecules. ICLR, 2023.

---

> > > > > > > > ### Author Response · Authors · 2024-12-01
> > > > > > > > **Response to **Reviewer DYuZ** (Part Ⅸ)**
> > > > > > > >
> > > > > > > > **2.** *This work would be solid research but incremental.*
> > > > > > > >
> > > > > > > > We are truly sorry to hear about your concerns, especially after we addressed many of them in our rebuttal. Nevertheless, we sincerely appreciate your valuable feedback and the positive recognition of our work. We believe you have gained a solid understanding of our approach, but please allow me to briefly reiterate the key contributions of our work.
> > > > > > > >
> > > > > > > > As you mentioned during the review stage, *This method is based on two main ideas: (1) identifying subgraph pairs (core substructures) that contribute most to prediction accuracy, based on Graph Information Bottleneck (GIB) theory (Yu et al., 2020; 2022b), and (2) implementing out-of-distribution generalization through invariant learning (Wu et al., 2022a) to mitigate environmental factors that may disrupt estimation.* Therefore, the novelty of I2Mole lies not in any individual component—whether the GIB theory or the use of the VQ module—but in its unified framework.
> > > > > > > >
> > > > > > > > In the field of molecular relational learning, **modeling molecular interactions remains underdeveloped**. Existing methods are proficient at elucidating essential structural features for individual molecules, but when molecular interactions occur, crucial substructures may exhibit substantial variation. To address this, we introduced the concept of a merged graph to collectively extract substructures via the Information Bottleneck (GIB) theory. **Furthermore, considering the vast and largely unexplored chemical space and the diversity of molecular combinations, we leverage invariant learning to enhance model generalization**. Specifically, as you noted, *several latent environmental factors in the training data are represented as learnable embedding vectors, and prediction robustness is enhanced by clustering non-core substructure variations through learning vector quantization (LVQ).* Based on this framework, our model demonstrates superior predictive accuracy and generalization compared to established approaches in drug-drug interaction (DDI) tasks.
> > > > > > > >
> > > > > > > > Technically, we effectively integrate GIB and VQ within the framework of invariant learning theory, significantly enhancing the model's performance. On the other hand, while the AI4Science community has made considerable progress in molecular-related work, molecular relational learning remains a crucial area of research, especially in fields like chemistry and drug discovery. This is because most cellular functions or drug effects rely on the joint action of multiple molecules or proteins. Our goal is to fill the current gap in modeling molecular interactions and improving model generalization, encouraging further focus from the AI4Science community on this task and thereby advancing the resolution of a major bottleneck in drug discovery.
> > > > > > > >
> > > > > > > > --------
> > > > > > > > **Thank you once again for your thorough, insightful, and constructive reviews. We sincerely appreciate your valuable feedback, and we have made every effort to address the concerns you raised. If our responses have satisfactorily addressed your concerns, we would respectfully hope you might consider granting a positive score to support our work. We would be truly grateful for your consideration.**

---

### Official Review · Reviewer_8jyg · 2024-11-03

**Soundness:** 3
**Presentation:** 3
**Contribution:** 3
**Rating:** 6
**Confidence:** 3

**Summary:**

The manuscript proposes a novel framework for generalizable molecular property prediction by enhancing generalizability using graph information bottleneck theory and an environment codebook. Experimental results show that the proposed method outperforms baseline methods on three benchmark datasets.

**Strengths:**

1. The method is presented clearly.

2. The proposed approach demonstrates superior performance and generalizability on benchmark datasets.

**Weaknesses:**

1. The method has not been evaluated on common DDI benchmarks, such as those referenced in "Comprehensive Evaluation of Deep and Graph Learning on Drug–Drug Interaction Prediction".

2. It is unclear why the information bottleneck (IB) approach improves performance. As shown in Table 5, IB significantly boosts model performance, raising AUROC from 87 to 94. Providing insights or explanations about the underlying mechanism for this improvement would help readers better understand its impact.

**Questions:**

1. What does “environment” represent chemicallly?

2. Could you provide examples illustrating how IB enhances model performance?

---

> ### Author Response · Authors · 2024-11-21
> **Response to **Reviewer  8jyg** (Part I)**
>
> Thank you for the valuable feedback. I will address each of your concerns in detail.
>
> >***W1:** The method has not been evaluated on common DDI benchmarks.*
>
> Thank you for your feedback. Following reference [1], we have incorporated two additional datasets, **DrugBank** and **Twosides**, to evaluate the performance of I2Mole (In Acc index).
>
>
> |            | **DrugBank**              |                        |                 | **Twosides**              |                        |                 |
> | ---------- | ------------------------- | ---------------------- | --------------- | ------------------------- | ---------------------- | --------------- |
> |            | **transductive settings** | **inductive settings** |                 | **transductive settings** | **inductive settings** |                 |
> |            |                           | **Type 1**             | **Type2**       |                           | **Type 1**             | **Type2**       |
> | MR-GNN     | 96.04(0.05)               | 74.67(0.33)            | 62.63(0.77)     | 76.23(0.23)               | 76.28(1.05)            | 63.25(0.81)     |
> | MHCADDI    | 83.80(0.27)               | 70.58(0.94)            | 66.50(0.62)     | -                         | 71.45(1.26)            | 68.53(0.57)     |
> | SSI-DDI    | 96.33(0.09)               | 76.38(0.92)            | 65.40(1.30)     | 78.20(0.14)               | 77.42(1.05)            | 67.43(1.52)     |
> | GAT-DDI    | 89.81(1.00)               | 69.83(1.41)            | 66.31(0.61)     | 50.00(0.20)               | 70.16(1.62)            | 68.57(0.24)     |
> | GMPNN-CS   | 95.30(0.05)               | 77.72(0.30)            | 68.57(0.30)     | 82.83(0.14)               | 78.56(0.51)            | 70.22(1.52)     |
> | SA-DDI     | 96.23(0.01)               | 75.55(1.12)        | 67.15(0.88) | 87.45(0.03)               | 78.32(0.78)            | 69.24(0.20)     |
> | DSN-DDI    | 96.94(0.02)           | 81.92(1.20)      | 73.42(1.29) | 98.83(0.04)               | 84.35(1.05)            | 74.26(1.25)     |
> | **I2Mole** | **97.25(0.04)**           | **82.43(1.23)**        | **74.32(1.42)** | **98.98(0.02)**           | **86.79(1.04)**        | **78.66(1.70)** |
>
> Clearly, I2Mole outperforms current baselines significantly on the **DrugBank** and **Twosides** datasets, with improvements of 0.319% and 0.15%, respectively, in the transductive setting.  More importantly, under **inductive settings**, I2Mole demonstrates even more substantial improvements compared to the second-best model. Specifically, it achieves gains of 0.622% and 2.89% in the **Type 1** scenario, and 1.22% and 5.9% in the **Type 2** scenario. These results highlight I2Mole's superior predictive accuracy and stronger generalization capability.
>
> In terms of this work, we utilize graph embedding-based techniques to acquire potentially effective molecular features. Then we evaluated on **ZhangDDI**, **ChchMiner**, and **DeepDDI**, which are also widely used benchmark datasets for molecular relationship learning [2,3,4]. Since the primary focus of this work is on the **out-of-distribution (OOD) generalization** of DDI tasks, we designed a series of experiments to assess model performance. First, we conducted evaluations under both **transductive settings** (where both molecules appear in the training set) and **inductive settings**:
>
> - **Type 1**: Predicts potential interaction properties between known and unseen drugs.
>
> - **Type 2**: Predicts potential interaction properties between unseen drugs.
>
> These settings not only evaluate the model's performance under conventional testing methodologies but also provide insights into its generalization ability to unseen molecules, a key aspect of OOD testing [2]. Further, we conducted generalization tests based on **scaffold splitting** and **size splitting**, similar to conventional single-molecule systems [5,6]. We also performed domain shift tests, following the design of **DSIL-DDI** [6], to validate the model's domain generalization performance. It is worth noting that the experimental setup for DDI tasks varies depending on the specific research focus. For example, **ZeroDDI** [8] focuses on zero-shot DDI event prediction, and **KnowDDI** [9] aims to uncover DDI relationships under long-tail distributions, where dataset splits are based on classes.
>
> Given our emphasis on **generalization**, our experimental setup differs from these approaches, as we aim to explore a different aspect of the DDI problem [10]. We hope this provides clarity regarding the reason behind our experimental design.
>
> [1] Comprehensive Evaluation of Deep and Graph Learning on Drug–Drug Interaction Prediction, Briefings in Bioinformatics, 2023
>
> [2] SSI–DDI: substructure–substructure interactions for drug–drug interaction prediction, Briefings in Bioinformatics, 2021
>
> [3] Conditional Graph Information Bottleneck for Molecular Relational Learning. ICML 2023

---

> > ### Author Response · Authors · 2024-11-21
> > **Response to **Reviewer 8jyg** (Part II)**
> >
> > [4] Shift-Robust Molecular Relational Learning with Causal Substructure. KDD, 2023
> >
> > [5] Learning Substructure Invariance for Out-of-Distribution Molecular Representations. NIPS, 2022
> >
> > [6] DrugOOD: Out-of-Distribution (OOD) Dataset Curator and Benchmark for AI-aided Drug Discovery. AAAI, 2023
> >
> > [7] DSIL-DDI: A Domain-Invariant Substructure Interaction Learning for Generalizable Drug-Drug Interaction Prediction. TNNLS, 2024.
> >
> > [8] ZeroDDI: A Zero-Shot Drug-Drug Interaction Event Prediction Method with Semantic Enhanced Learning and Dual-Modal Uniform Alignment. IJCAI, 2024
> >
> > [9] Accurate and interpretable drug-drug interaction prediction enabled by knowledge subgraph learning. Nature Communications Medicine, 2024
> >
> > [10] DSN-DDI: an accurate and generalized framework for drug–drug interaction prediction by dual-view representation learning. Briefings in Bioinformatics, 2023
> >
> > >***W2&Q2:** IB significantly boosts model performance, raising AUROC from 87 to 94. Providing insights or explanations about the underlying mechanism for this improvement would help readers better understand its impact.*
> >
> > Following your suggestion, we have included more interpretability results in the appendix to aid readers' understanding.
> >
> > The improvement in AUROC indicates that the model has become more precise in learning class boundaries, particularly in its ability to rank positive and negative samples with greater confidence. This advancement can be attributed to the Information Bottleneck (IB) principle, which extracts concise substructural features. By compressing redundant information while retaining key features, the model is better able to focus on core patterns relevant to classification, thereby enhancing its ability to distinguish boundary samples.
> >
> > Notably, for boundary samples that are challenging to classify due to noise interference, the features extracted by IB help the model overcome the limitations of traditional classification decision-making, enabling more accurate classification of these samples. This demonstrates that the model's capacity for feature selection and noise suppression has been significantly enhanced.
> >
> > > ***Q1:** What does “environment” represent chemically?*
> >
> > In this paper, "environment" refers to molecular contexts that influence properties and can vary across data distributions. These environments may be explicitly shaped by features such as molecular scaffolds or sizes, and they capture variations arising from molecular structure, experimental conditions, or biological contexts.
> >
> > For instance, consider formic acid (CH₂O₂) and citric acid (C₆H₈O₇): while these molecules differ in size and molecular scaffolds, they share common substructures, such as carboxy groups, which contribute to certain invariant properties like water solubility. In the context of water solubility, we treat the carboxy groups as rationales, whereas the remaining molecular scaffolds or functional groups are referred to as the "environment," as they influence the model's predictions.
> >
> > To enhance the model's ability to generalize to unseen samples, we construct diverse "environment" vectors to simulate molecular pairs, thereby improving the model's focus on the rationales and strengthening its generalization capabilities [1,2,3].
> >
> > [1] Shift-Robust Molecular Relational Learning with Causal Substructure. KDD, 2023
> >
> > [2] DSIL-DDI: A Domain-Invariant Substructure Interaction Learning for Generalizable Drug-Drug Interaction Prediction. TNNLS, 2024.
> >
> > [3] Learning Substructure Invariance for Out-of-Distribution Molecular Representations. NIPS, 2024
> >
> > > ***Q2:** Provide examples about the captured substructures.*
> >
> > Thank you for your valuable suggestion. In the revised manuscript, I have included the examples of substructure mining in the appendix. These cases demonstrate that I2Mole can effectively capture core substructures and indicate that intermolecular interactions are primarily attributable to interactions between specific substructures.
> >
> > ----------
> >
> > We greatly appreciate your insightful and helpful comments, as they will undoubtedly help us improve the quality of our article. If our response has successfully addressed your concerns and clarified any ambiguities, we respectfully hope that you consider raising the score. Should you have any further questions or require additional clarification, we would be delighted to engage in further discussion. Once again, we sincerely appreciate your time and effort in reviewing our manuscript. Your feedback has been invaluable in improving our research.

---

> > > ### Comment · Reviewer_8jyg · 2024-11-23
> > > **Response**
> > >
> > > Thanks for the response. I will remain my score.

---

> > > > ### Author Response · Authors · 2024-11-25
> > > > **Response to  Reviewer 8jyg**
> > > >
> > > > Dear **Reviewer 8jyg**,
> > > >
> > > > Thank you for your positive feedback and continued support for our paper! We appreciate your thoughtful review and are glad that we have adequately addressed your concerns.
> > > >
> > > > Best regards,
> > > >
> > > > Authors.

---

### Official Review · Reviewer_b1KJ · 2024-11-03

**Soundness:** 2
**Presentation:** 2
**Contribution:** 2
**Rating:** 6
**Confidence:** 4

**Summary:**

This paper presents a framework, I2Mole (Interaction-aware Invariant Molecular Learning), that captures atomic interactions by establishing broad connections between intermolecular atoms and refining them through an improved graph information bottleneck theory. This approach can capture core substructures critical to interactions and, in addition, create an environment codebook of the merged graph, which enhances the generalizability and preserves chemical semantics. Extensive experiments were carried out to validate the proposed approach.

**Strengths:**

* Combines Graph Information Bottleneck and Invariant Learning to tackle molecular interaction issues.
* Extensive experiments that yield interesting findings.

**Weaknesses:**

* Some of the math notations are confusing. For example, but not limited to the following. In equation (1), G_IB on the left side should be different from that on the right side. Similar confusion appears in equation (12). Equation (2) has a typo (it seems that the expectation term is put in the subscript), and "env" and "E" are not clearly defined.

* The construction of core reaction substructure candidate in the merged graph can be better explained. Section 3.2.2 explains see some of the components such as zi (updated node representation with injection of noise) and h_i^r (irrelevant substructure node). More clarification is needed to explain how to combine these components to construct $g^ ̃_IB$ in details.

* Regarding code embedding of environment, it is unclear how to decide M. For example, what criteria and/or method(s) can be used to justify a choice of M.

**Questions:**

See the weakness.

---

> ### Author Response · Authors · 2024-11-21
> **Response to **Reviewer b1KJ****
>
> Thank you for the valuable feedback. I will address each of your concerns in detail.
> > ***W1**: Some of the math notations are confusing in equation (1), (2) and (12). And "env" and "E" are not clearly defined.*
>
> Thank you for your feedback. We have adopted a more standardized format to redefine Equation (1) and Equation (12). These changes have been incorporated into the revised version for clarity and consistency.
>
> Equation (1):
>
> $$
> \mathcal{G} _ {\mathrm{IB}} = \underset{\mathcal{G} _ {\mathrm{sub}} \in \mathcal{S}}{\arg\min} \, -I(\mathbf{Y}; \mathcal{G} _ {\mathrm{sub}}) + \beta I(\mathcal{G}; \mathcal{G} _ {\mathrm{sub}}).
> $$
>
> Intuitively, where $\mathcal{S}$ represents the set of $\mathcal{G} _ {\mathrm{sub}}$, $\mathcal{G} _ {\mathrm{sub}}$ is the general subgraph.
>
>
> Equation (12):
>
> $$
> \mathcal{\widetilde{G}} _ {\mathrm{IB}} = \underset{\mathcal{\widetilde{G}} _ {\mathrm{sub}} \in \mathcal{\widetilde{S}}}{\arg\min} \, -I(\mathbf{Y}; \mathcal{\widetilde{G}} _ {\mathrm{sub}}) + \beta I(\mathcal{G}; \mathcal{\widetilde{G}} _ {\mathrm{sub}}),
> $$
>
> where $\mathcal{\widetilde{S}}$  represents the set of $\mathcal{\widetilde{G}} _ {\mathrm{sub}}$. Each term indicates the prediction and compression terms respectively, which should be minimized during training, as outlined below.
>
> Equation (2):
>
> $$
> \min _ f \max _ {{\mathcal{G}_{\text{env}}} \in \mathbf{E}} \mathbb{E} _ {(\mathcal{G}, Y) \sim p(\mathcal{G}, \mathbf{Y} \mid \mathbf{env}={\mathcal{G} _ {\text{env}})}}[R(f(\mathcal{G}), \mathbf{Y}) \mid {\mathcal{G} _ {\textbf{env}}]}
> $$
>
> Here, $\mathbf{E}$ represents the *set of environments*, i.e., the collection of all possible environments $\mathcal{G} _ {\text{env}}$. $\mathcal{G} _ {\text{env}}$ is a specific instance of an environment, $env$ is an environment variable in environment codebook $W$, and $\mathbb{E}$ represents the mathematical expectation.
>
> > ***W2**:  More clarification is needed to explain how to combine these components to construct $\mathcal{\widetilde{G}} _ {\mathrm{IB}}$ in details.*
>
> We sincerely apologize for the writing error. The correct statement should be: $\mathbf{h}^{r} _ {i}$ is the irrelevant substructure node which would be used to construct $\mathcal{\widetilde{G}} _ {\mathrm{env}}$ (in section 3.3). Here, $h _ {i}$ denotes the original node feature, and $\lambda _ {i}$ is the weight assigned to the node.
>
> During the construction of node embeddings $z _ {i}$ in $\mathcal{\widetilde{G}} _ {\mathrm{IB}}$, it is formed by the combination of $\lambda _ {i}h _ {i}$ and $(1-\lambda _ {i})\epsilon$, where $\lambda_ {i}h _ {i}$ represents the rationale information, i.e., the core features. In this work, $(1-\lambda_ {i})\epsilon$ is considered as the environmental feature. This environmental feature originates from the molecule itself but lacks robustness. To address this, we subsequently leverage $\mathbf{h}^{r} _ {i}$ to identify new environments and construct the final representations, as described in **Section 3.3**.
> $$
> z _ {i} = \lambda _ {i}h _ {i} + (1-\lambda _ {i})\epsilon
> $$
>
> > ***W3**: How to decide M.*
>
> $M$ represents the environmental vector number in the environment codebook, which is treated as a hyperparameter, as recorded in Table 8. We have conducted hyperparameter experiments in original version, as presented in Figure 6, to determine the optimal value for $M$.
>
> ------------
>
> We greatly appreciate your insightful and helpful comments, as they will undoubtedly help us improve the quality of our article. If our response has successfully addressed your concerns and clarified any ambiguities, we respectfully hope that you consider raising the score. Should you have any further questions or require additional clarification, we would be delighted to engage in further discussion. Once again, we sincerely appreciate your time and effort in reviewing our manuscript. Your feedback has been invaluable in improving our research.

---

> > ### Comment · Reviewer_b1KJ · 2024-11-23
> > **Read the responses**
> >
> > Updated the score to 6

---

> ### Author Response · Authors · 2024-11-25
> **Response to Reviewer b1KJ**
>
> Dear **Reviewer b1KJ**,
>
> Thank you for your positive feedback and continued support for our paper! We appreciate your thoughtful review and are glad that we have adequately addressed your concerns.
>
> Best regards,
>
> Authors.

---

### Official Review · Reviewer_RyBk · 2024-11-04

**Soundness:** 3
**Presentation:** 2
**Contribution:** 3
**Rating:** 6
**Confidence:** 4

**Summary:**

The motivation for this paper stems from limitations in molecular interaction modeling and a lack of focus on out-of-distribution (OOD) generalization capability. To address these challenges, the authors introduce I2Mole, a method designed to enhance both molecular interaction modeling and OOD generalization.

To capture emergent properties in molecular interactions—characteristics that do not appear in independent molecules—the approach constructs a merged graph that connects the interacting molecule pair, leveraging the Graph Information Bottleneck (GIB) to extract the rational.

Subsequently, the model utilizes a codebook to enhance OOD capabilities, ensuring consistency across various environments. This codebook enables the model to adapt more effectively to diverse conditions, thereby improving generalization performance in unfamiliar scenarios.

**Strengths:**

* The paper clearly explains its key claims, including limitations in molecular interaction modeling and the lack of focus on out-of-distribution (OOD) generalization.

* The proposed method is technically sound. Constructing a merged graph that combines two molecules to model molecular interactions and using the Graph Information Bottleneck (GIB) to identify important substructures is an intuitive and effective approach. Additionally, the introduction of a VQ codebook to account for diverse environments is a simple yet effective strategy to enhance OOD generalization.

* The authors conducted comprehensive experiments across multiple datasets, demonstrating improved performance compared to baseline methods.

**Weaknesses:**

* More detailed explanations are needed regarding the necessity of each component. For instance, it would be helpful to clarify: 1) why intra-molecular and inter-molecular message passing need to be separated after constructing the merged graph, and 2) while the limitations of simulating diverse environmental distributions through noise injection are well discussed, further elaboration on the role and benefits of vector quantization in this context would enhance understanding.

* There is insufficient consideration of environmental factors. The environment is defined based on portions of the merged graph outside of the core substructure, yet the physical and chemical environment where interactions occur, as well as other external conditions, are not considered.

* Although quantitative metrics demonstrate the model’s superiority, it is disappointing that the paper does not provide evidence that the model captures emergent structures specific to interactions, as suggested in the introduction. Showing such examples would better highlight the effectiveness of the model in learning interaction-specific substructures.

* Numerous formatting, notation, and equation issues need correction:
    * Inconsistent usage of "I2Mol" and "I2Mole."
    * In Equation (5), the definition of $k$ is missing
    * Redundant wording in "$u$ is is updated into $u_i^{'}$."
    * Misaligned square brackets in Equation (6).
    * In Equation (10), $s$ is defined by the top $s$%, but the equation appears to represent a threshold.
    * In Equation (24), the definition of $||$ is missing.
    * The following sentence in Section 4.2 seems to belong in Section 4.3: "Type 1 aims to predict potential interaction properties between known and unseen drugs, while Type 2 aims to predict potential interaction properties between unseen and unseen drugs."

**Questions:**

* Could you provide more detailed explanations on why each component is necessary? For example, why do intra-molecular and inter-molecular message passing need to be separated after constructing the merged graph? Also, what are the role and benefits of vector quantization in simulating diverse environmental distributions? (Related to W1)
* Can you provide examples showing that the model captures emergent structures specific to interactions, as mentioned in the introduction? (Related to W3)

---

> ### Author Response · Authors · 2024-11-21
> **Response to **Reviewer RyBk** (Part I)**
>
> Thank you for the valuable feedback. I will address each of your concerns in detail.
>
> > ***W1&Q1:** why intra-molecular and inter-molecular message passing need to be separated after constructing the merged graph*
>
> Thank you very much for your insightful suggestion. The separation of intra-molecular and inter-molecular message passing is motivated by the distinct physical and chemical characteristics of these interactions. Intra-molecular interactions, such as covalent bonds and specific geometric constraints, often adhere to well-defined physical rules, whereas inter-molecular interactions involve weaker forces like hydrogen bonds, van der Waals forces, and other non-covalent interactions [1,2].
>
> Handling these two types of message passing separately allows us to better capture their unique properties. Moreover, this separation enables the design of tailored optimization strategies or network architectures (e.g., differing retention rates for relational edges) to efficiently model each type of interaction. For example, in our approach, we employed MPNN for intra-molecular message passing to effectively model the structured, rigid interactions within molecules, while using TGAT for inter-molecular message passing to account for the transient and weaker nature of inter-molecular interactions.
>
> This division is particularly important because integrating these two types of interactions without distinction could introduce unwanted noise. Specifically, insufficient incorporation of inter-molecular information could weaken the learning of inter-molecular interactions, while excessive inclusion might disrupt the intrinsic properties of intra-molecular relationships, as demonstrated in Table 10. Additionally, handling an excessive number of relational edges simultaneously could unnecessarily increase model complexity.
>
> Considering these factors, we concluded that separating intra-molecular and inter-molecular message passing is essential for effectively capturing their respective characteristics and improving model efficiency.
>
> In revised version, we have added relevant descriptions in the **Introduction section** to facilitate readers' understanding.
>
> [1] Chi Z. Toxic interaction mechanism between oxytetracycline and bovine hemoglobin. J. Hazard, 2010
>
> [2] Chen D. Algebraic graph-assisted bidirectional transformers for molecular property prediction. Nat.Commun, 2021.
>
> >***W1&Q1:** Further elaboration on the role and benefits of vector quantization in this context.*
>
> Thank you for your thoughtful suggestion. As mentioned in the **Introduction**, randomly simulated noise and indiscriminate noise injection can introduce potential issues, such as instability and lack of chemical interpretability. To address these concerns, we leveraged the non-core substructures of the merged molecular graph $\widetilde{\mathcal{G}}_\text{env}$ as latent environmental information. This adjustment not only enhances model robustness and generalization but also aligns with the underlying chemical properties of the data.
>
> Given the vast and largely unknown nature of the potential chemical space, collecting all possible environments is impractical. Therefore, the VQ module clusters an infinite number of potential environments into a finite set. This clustering enables us to approximate a diverse range of environment, derived from pooling results of less important nodes (Eq. 22), thus providing a more computationally feasible and chemically meaningful representation.
>
> To further illustrate the role of the VQ module, we introduced extra two variants for comparison in this revised version:
>
> 1. **RD Noise Variant**: In this version, noise in the environment codebook is entirely random, mimicking the effects of random noise injection.
>
> 2. **Instance-Dependent (ID) Noise Variant**: Here, we sampled new environments from the environment codebook and added them as small perturbations to the instance-dependent environment. The Gaussian distribution of this noise is determined by the mean and variance of subgraph node vectors, emphasizing instance-dependent noisy perturbations.
>
>
> |              | **ZhangDDI** |              | ChchMiner    |              | DeepDDI      |              |
> | ------------ | ------------ | ------------ | ------------ | ------------ | ------------ | ------------ |
> | Method       | Acc (↑)      | AUROC (↑)    | Acc (↑)      | AUROC (↑)    | Acc (↑)      | AUROC (↑)    |
> | **RD noise** | 87.21 (0.11) | 93.76 (0.13) | 93.47 (0.08) | 97.52 (0.07) | 92.39 (0.38) | 97.01 (0.39) |
> | **ID noise** | 88.02 (0.06) | 94.47 (0.08) | 94.27 (0.12) | 98.54 (0.06) | 94.56 (0.10) | 97.42 (0.31) |
> | **Ours**     | 88.64 (0.24) | 95.12 (0.12) | 95.34 (0.19) | 98.84 (0.10) | 96.51 (0.14) | 99.04 (0.22) |
>
>
> Our experimental results demonstrate the superiority of the VQ module and the proposed optimization strategy, particularly in improving the model's robustness and ability to generalize across diverse chemical environments.

---

> > ### Author Response · Authors · 2024-11-21
> > **Response to **Reviewer RyBk** (Part II)**
> >
> > >***W1&Q1**: The physical and chemical environment where interactions occur, as well as other external conditions, are not considered.*
> >
> > Thank you for your insightful suggestion. This is indeed a crucial issue and aligns with one of our ongoing research directions. Currently, due to the limited availability of datasets and the relatively low level of attention this type of problem has received within the community, most existing test scenarios do not account for physical conditions or external factors [1, 2, 3]. Therefore, we have made a discussion in the **Limitation section**: *“An equally important aspect is that drugs often function only under specific conditions such as temperature and pH levels…”* (**Appendix F**).
> >
> > We are aware of some research efforts that have begun exploring the impact of different environmental conditions on target systems, such as Tapioca [4] and HMNN [5]. Building upon these studies, we are considering integrating multimodal information, incorporating additional feature representations, and leveraging multi-expert model ensembles. These approaches hold significant potential for improving model performance in addressing such challenges. We believe this line of inquiry will be both intellectually stimulating and impactful, and we aim to pursue it as a key direction in our future work.
> >
> > [1] Conditional Graph Information Bottleneck for Molecular Relational Learning, 2023 ICML
> >
> > [2] Comprehensive Evaluation of Deep and Graph Learning on Drug–Drug Interaction Prediction, 2023, Brief Bioinform.
> >
> > [3] Shift-Robust Molecular Relational Learning with Causal Substructure，2023 KDD
> >
> > [4] Tapioca: a platform for predicting de novo protein–protein interactions in dynamic contexts. 2024 Nat Methods
> >
> > [5] Spectroscopy-Guided Deep Learning Predicts Solid–Liquid Surface Adsorbate Properties in Unseen Solvents. 2024 JACS
> >
> > > ***W3&Q2**: Can you provide examples showing that the model captures emergent structures?*
> >
> > Thank you for your valuable suggestion. In the revised manuscript, I have included the examples of substructure mining in the appendix. These cases demonstrate that I2Mole can effectively capture core substructures and indicate that intermolecular interactions are primarily attributable to interactions between specific substructures.
> >
> > > ***W4**:Numerous formatting, notation, and equation issues need correction.*
> >
> > * Inconsistent usage of "I2Mol" and "I2Mole."
> >
> > Thank you for your careful review. In the revised version, we have standardized the term to "I2Mole" throughout the manuscript and highlighted the changes for easier verification.
> >
> > * In Equation (5), the definition of k is missing
> >
> > Thank you for your careful review. Here, $ k $ refers to the index of the relational edge $R$, which we have clarified in the revised version.
> >
> > * Redundant wording in "u is is updated into ui′."
> >
> > Thank you for your careful review. We have removed the redundant "is" in the revised version.
> >
> > * Misaligned square brackets in Equation (6).
> >
> > We sincerely apologize for the oversight. We have revised the equation to:
> >
> > $$
> > e _ {ij}^{\prime} = e _ {ij} + {\rm LeakyReLU}[{\rm FC}(v _ i + v _ j) + [{\rm FC}(e _ {ij}) + [{\rm FC}(u)]],
> > $$
> >
> > and ensured that the square brackets are properly aligned in the revised version. Thank you for pointing this out.
> >
> > * In Equation (10), s is defined by the top s%, but the equation appears to represent a threshold.
> >
> > Indeed, this refers to retaining the top $s\%$ of relational edges after weighting and sorting. In the revised version, we have revised this description for further clarification and corrections. "*Here, s represents the threshold corresponding to the $top\_s\%$ ranking of $r _ {ij}$ values. The selected attention coefficients are then normalized across the entire graph to facilitate the intermolecular information-passing process.*"
> >
> > * In Equation (24), the definition of $||$ is missing.
> >
> > Thank you for your careful review. Here, $||$ refers to the concatenation operation, and we have provided additional clarification in the revised version.
> >
> > * The following sentence in Section 4.2 seems to belong in Section 4.3: "Type 1 aims to predict potential interaction properties between known and unseen drugs, while Type 2 aims to predict potential interaction properties between unseen and unseen drugs."
> >
> > Thank you for your professional feedback. You are absolutely correct that this sentence was misplaced in **Section 4.2**. We have now moved it to **Section 4.3** in the revised version.
> >
> > ------------
> >
> > We greatly appreciate your insightful and helpful comments, as they will undoubtedly help us improve the quality of our article. If our response has successfully addressed your concerns and clarified any ambiguities, we respectfully hope that you consider raising the score. Should you have any further questions or require additional clarification, we would be delighted to engage in further discussion. Once again, we sincerely appreciate your time and effort in reviewing our manuscript.

---

> > > ### Comment · Reviewer_RyBk · 2024-11-25
> > >
> > > Thank you for the thorough and well-explained response—it effectively addressed all my concerns. I’ve decided to increase my score. I have no further questions

---

> > > > ### Author Response · Authors · 2024-11-25
> > > > **Response to Reviewer RyBk**
> > > >
> > > > Dear Reviewer RyBk,
> > > >
> > > > Thank you for your positive feedback and continued support for our paper! We appreciate your thoughtful review and are glad that we have adequately addressed your concerns.
> > > >
> > > > Best regards,
> > > >
> > > > Authors.

---

### Author Response · Authors · 2024-11-21
**# General Response to All the Reviewers**

We thank all the reviewers for their insightful and constructive reviews of our manuscript. We are encouraged to hear that the reviewers find that **our idea is novel or interesting** ($\frac{3}{4}$Reviewers RyBk, 8jyg, DYuZ); **our motivation/research problem is clear/explicit** (All Reviewers). Also, they think that **our experimental results are comprehensive or promising** (All Reviewers); **Our work is well-presented and easy to follow** ($\frac{3}{4}$Reviewers RyBk, 8jyg, DYuZ).

We have carefully reviewed all the suggestions and provided detailed responses to each of the points. If there are any further questions or concerns, please don’t hesitate to let us know. We are actively engaged in the discussion and are committed to improving the quality of this work. Thanks once again for the valuable input!

---

### Meta-Review · Area_Chair_NdqW · 2024-12-20

**Metareview:**

This paper proposes I2Mole, a method designed to enhance molecular interaction modeling and out-of-distribution (OOD) generalization by constructing a merged graph and leveraging the Graph Information Bottleneck (GIB) to identify core substructures. The method also employs a codebook to improve OOD generalization across diverse environments. However, the issues raised by the reviews are critical, especially the clarity issues and the possible misleading statement.

**Additional Comments On Reviewer Discussion:**

Although the paper has some merits, such as its intuitive approach to modeling molecular interactions and the use of a VQ codebook for OOD generalization, the issues raised by the reviews are critical, especially the clarity issues and the possible misleading statement. Although the authors address some issues in their responses, the paper still needs a major revision before it can be accepted.

---

### Decision · Program_Chairs · 2025-01-22

Reject